# Intrinsic motivation for choice varies with individual risk attitudes and the controllability of the environment

**Jérôme Munuera**[1,2]*, **Marta Ribes Agost**[2], **David Bendetowicz**[1], **Adrien Kerebel**[2], **Valérian Chambon**[2‡]*, **Brian Lau**[1‡]*

**1** Sorbonne Université, Institut du Cerveau—Paris Brain Institute—ICM, Inserm, CNRS, APHP, Paris, France, **2** Institut Jean Nicod, Département d'études cognitives, ENS, EHESS, CNRS, PSL University, Paris, France

‡ These authors share last authorship on this work.
* jerome.munuera@icm-institute.org (JM); valerian.chambon@ens.fr (VC); brian.lau@upmc.fr (BL)

**Data Availability Statement:** Data underlying the findings reported is publicly available (DOI: 10.6084/m9.figshare.22269625). Scripts underlying the findings reported are included with the data

## Abstract

When deciding between options that do or do not lead to future choices, humans often choose to choose. We studied choice seeking by asking subjects to first decide between a choice opportunity or performing a computer-selected action, after which they either chose freely or performed the forced action. Subjects preferred choice when these options were equally rewarded, even deterministically, and traded extrinsic rewards for opportunities to choose. We explained individual variability in choice seeking using reinforcement learning models incorporating risk sensitivity and overvaluation of rewards obtained through choice. Model fits revealed that 28% of subjects were sensitive to the worst possible outcome associated with free choice, and this pessimism reduced their choice preference with increasing risk. Moreover, outcome overvaluation was necessary to explain patterns of individual choice preference across levels of risk. We also manipulated the degree to which subjects controlled stimulus outcomes. We found that degrading coherence between their actions and stimulus outcomes diminished choice preference following forced actions, although willingness to repeat selection of choice opportunities remained high. When subjects chose freely during these repeats, they were sensitive to rewards when actions were controllable but ignored outcomes–even positive ones–associated with reduced controllability. Our results show that preference for choice can be modulated by extrinsic reward properties including reward probability and risk as well as by controllability of the environment.

## Author summary

Human decisions can often be explained by the balancing of potential rewards and punishments. However, some research suggests that humans also prefer opportunities to choose, even when these have no impact on future rewards or punishments. Thus, opportunities to choose may be intrinsically motivating, although this has never been experimentally tested against alternative explanations such as cognitive dissonance or

(DOI: 10.6084/m9.figshare.22269625). The R package with the code we wrote for simulating and fitting reinforcement learning models is freely available under the MIT open-source license (DOI: 10.6084/m9.figshare.22340983).

**Funding:** J.M,V.C and B.L. were supported by the Agence Nationale de la Recherche (ANR, https://anr.fr/) grant ANR-19-CE37-0014-01 (ANR PRC) and by the European Commission (https://ec.europa.eu/, H2020-MSCA-IF-2018-#845176 to J.M.). D.B. was supported by a Fondation pour la Recherche Médicale (FRM, https://www.frm.org/) fellowship (FDM201906008526). V.C. was supported by the ANR grants ANR-17-EURE-0017 (Frontiers in Cognition), ANR-10-IDEX-0001-02 PSL (program 'Investissements d'Avenir'), ANR-16-CE37-0012-01 (ANR JCJ). The funders had no role in study design, data collection and analysis, decision to publish, or preparation of the manuscript.

**Competing interests:** The authors have declared that no competing interests exist.

exploration. We conducted behavioral experiments and used computational modelling to provide compelling evidence that choice opportunities are indeed intrinsically rewarding. Moreover, we found that human choice preference can compete with maximizing reward and can vary according to individual risk attitudes and the controllability of the environment.

## Introduction

Preference for choice has been observed in humans [1–7] as well as other animals including rats [8], pigeons [9] and monkeys [10,11]. This free-choice premium can be behaviorally measured by having subjects perform trials in two stages: a decision is first made between the opportunity to choose from two terminal actions (*free*) or to perform a mandatory terminal action (*forced*) in the second stage [8]. Although food or fluid rewards follow terminal actions in non-human studies, choice preference in humans can be elicited using hypothetical outcomes that are never obtained [3,12]. Thus, choice opportunities appear to possess or acquire value in and of themselves. It may be that choice has value because it represents an opportunity to exercise control [13], which is itself intrinsically rewarding [1,4,14]. Personal control is central in numerous psychological theories, where constructs such as autonomy [15,16], controllability [17,18], personal causation [19], effectance [20], perceived behavioral control [21] or self-efficacy [17] are key for motivating behaviors that are not economically rational or easily explained as satisfying basic drives such as hunger, thirst, sex, or pain avoidance [22].

There are alternative explanations for choice seeking. For example, subjects may prefer choice because they are curious and seek information [23,24], or they wish to explore potential outcomes to eventually exploit their options [25], or because they seek variety to perhaps reduce boredom [26] or keep their options open [3]. By these accounts, however, the expression of personal control is not itself the ends, but rather a means for achieving an objective that once satisfied reduces choice preference. For example, choice preference should decline when there is no further information to discover in the environment, or after uncertainty about reward contingencies have been satisfactorily resolved.

Choice seeking may also arise due to selection itself altering outcome representations. Contexts signaling choice opportunities may acquire distorted value through choice-induced preference change [27]. By this account, deciding between equally valued terminal actions generates cognitive dissonance that is resolved by post-choice revaluation favoring the chosen action [27,28]. This would render the free option more valuable than the forced option since revaluation only occurs for self-determined actions [29,30]. Alternatively, subjects may develop distorted outcome representations through a process related to the winner's or optimizer's curse [31], whereby optimization-based selection upwardly biases value estimates for the chosen action. One algorithm subject to this bias is Q-learning [32], where action values are updated using the maximum value to approximate the maximum expected value. In the two-stage task described above, the free action value is biased upwards due to considering only the best of two possible future actions, while the forced action value remains unbiased since there is only one possible outcome [33]. Again, the expression of personal control is not itself the ends for these selection-based accounts, and both predict that choice preference should be reduced when terminal rewards associated with the free option are clearly different.

Data from prior studies does not arbitrate between competing explanations for choice-seeking. Here, we used behavioral manipulations and computational modelling to explore the factors governing human preference for choice. In the first experiment, we altered the reward

contingencies associated with terminal actions in order to rule out curiosity, exploration, variety-seeking, and selection-based accounts as general explanations for choice seeking. In the second experiment, we assessed the value of choice by progressively increasing the relative value between trials with and without choice opportunity. We then used reinforcement learning models to show that optimistic learning (considering the best possible future outcome) was insufficient to explain individual variability in choice seeking. Rather, subjects adopted different *decision attitudes*, the desire to make or avoid decisions independent of the outcomes [12], which were balanced against differing levels of risk sensitivity. Finally, in the third experiment, we sought to test whether choice preference was modulated by control beliefs. Manipulating the controllability of the task–that is, the objective controllability over stimulus outcome–did not reduce the high willingness to repeat a free choice. However, subjects were sensitive to past rewards only in controllable trials, where stimulus outcomes could be attributed to self-determined choice. In contrast, during uncontrollable trials subjects ignored rewards and repeated their previous choice. We suggest that choice repetition in the face of uncontrollability reflects a strategy to compensate for reduced control over the environment, consistent with the broader psychology of control maintenance. Together, our results show that human preference for free choice depends on a trade-off between subjects' decision attitudes (overvaluation of reward outcome), risk attitudes, and the controllability of the environment.

## Results

Subjects performed repeated trials with a two-stage structure (Fig 1). In each trial, subjects made a 1st-stage choice between two options defining the 2nd-stage: the opportunity to choose between two fractal targets (*free*) or performing an obligatory selection of another fractal target (*forced*). Extrinsic rewards (€) were delivered only for terminal (i.e., 2nd-stage) actions. If subjects chose the *forced* option, the computer always selected the same fractal target for the subjects. If subjects chose the *free* option, they had to choose between two fractal targets associated with two different terminal states. We fixed reward contingencies in blocks of trials and used unique fractal targets for each block. We divided each block into an initial training phase with the same number of trials in *free* and *forced* options (Fig 1B; e.g., 48 trials for both *free* and *forced* trials, see Materials and Methods) followed by a test phase (Fig 1C) to ensure that the subjects learned the associations between the different fractal targets and extrinsic reward probabilities. Subjects were not told the actual extrinsic reward probabilities but were informed that reward contingencies did not change between the train and test phases.

### Free choice preference across different extrinsic reward probabilities

In experiment 1 (*n* = 58 subjects), we varied the overall expected value by varying the probability of extrinsic reward delivery (P) across different blocks of trials. These probabilities ranged from 0.5 to 1 across the blocks (i.e., low to high), and the programmed probabilities in *free* and *forced* 2nd-stage rewards were equal (Fig 2A). For example, in high probability blocks, we set the probabilities of the *forced* terminal action and of one of the *free* terminal actions (a1) to 1 and set the probability of the second *free* terminal action (a2) to 0. Therefore, the maximum expected value was equal for the *free* and *forced* options.

Subjects chose to choose more frequently, selecting the *free* option in 64% of test trials on average (Fig 2B). The level of preference did not differ significantly across blocks (low = 65%, medium = 62%, high = 66%; $\chi^2$ = 4.49, *p* = 0.106; see Fig A in S1 Text for individual subject data). We also examined 1st-stage reaction times (RT), which were not significantly different across different reward probabilities (estimated trend = 0.054, 95% CI = [-0.062, 0.175],

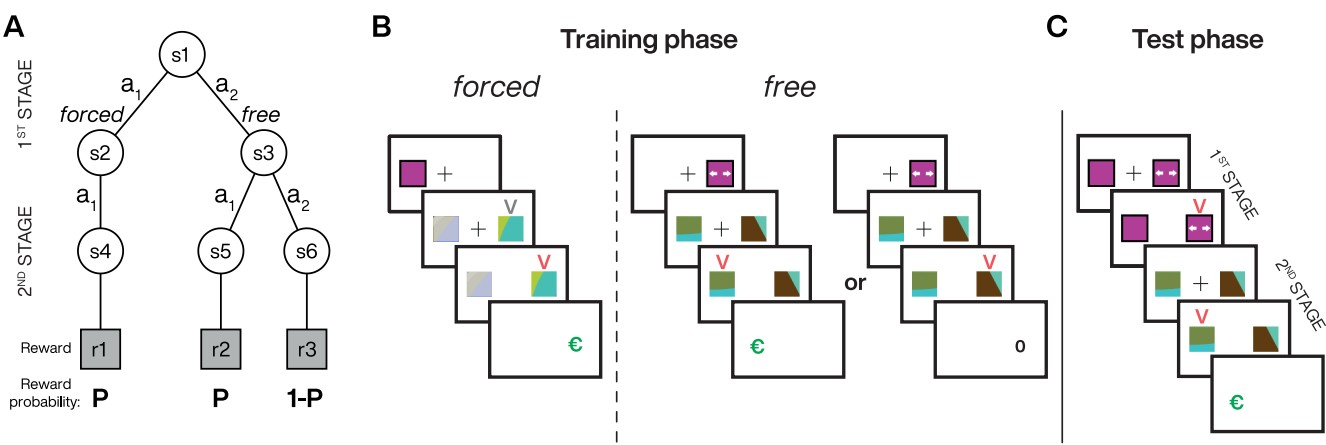

**Fig 1. Two-stage task structure. A.** State diagram illustrating the 6 possible states (s), actions (a) and associated extrinsic reward probabilities (e.g., P = 0.5, 0.75 or 1 for blocks 1 to 3, respectively); s2 and s3 were represented by two different 1st-stage targets (e.g., colored squares with or without arrows for *free* and *forced* trials, respectively) and s4 to s6 were associated to three different 2nd-stage targets (fractals). **B.** Sequence of events during the training phase where the subjects experienced the *free* or *forced* target at the 1st-stage, then learned the contingencies between the fractal targets and their reward probabilities at the 2nd-stage (P) associated with the *forced* (no choice) and *free* (choice available) options. When training the reward contingencies associated with the *forced* option, subjects' actions in the 2nd-stage had to match the target indicated by a grey V-shape, which always indicated the same fractal (s4). When training the reward contingencies associated with the *free* option, no mandatory target is present at the 2nd-stage (s5 or s6 can be chosen) but one of the targets is more rewarded when P > 0.5. **C.** Sequence of events during the test phase: subjects first decided between the *free* or *forced* option and then experienced the associated 2nd-stage. Rewards, when delivered, were represented by a large green euro symbol (€). At each stage of the task, a red V-shape appeared to indicate the target selected by the subjects either in *free* or *forced* trials.

$p$ = 0.370; Fig B(A) in S1 Text). We found that subjects immediately expressed above chance preference for the *free* option (Fig 2C) despite never having actualized 1st-stage choices during training. Looking within a block, we found that subjects' preference remained constant across trials in medium and high reward probability blocks ($\chi^2$ = 0.7, $p$ = 0.215 and $\chi^2$ = 0, $p$ = 0.664 for nonlinear smooth by trial deviating from a flat line, respectively; Fig 2C, middle and right panels). In low probability blocks, subjects started with a lower choice preference that gradually increased to match that observed in the medium and high probability blocks ($\chi^2$ = 13.2, $p$ = 0.001 for nonlinear smooth by trial; Fig 2C left panel). The lower reward probability may have prevented subjects from developing accurate reward representations by the end of the training phase, which may have led to additional sampling of the three 2nd-stage targets (two in *free* and one in *forced*) in the beginning of the test phase.

## Second-stage performance following *free* selection

We investigated participants' 2nd-stage choices following *free* selection to exclude the possibility that choice preference arose because reward contingencies had not been learned. During the training phase, when P>0.5, participants quickly learned to choose the most rewarded fractal targets (at P = 0.5, all fractal targets were equally rewarded) (Fig 2D). During the test phase, participants continued to select the same targets (Fig 2E), confirming stable application of learned contingencies ($p$ > 0.1 for nonlinear smooth by trial deviating from a flat line for all blocks).

Choice preference was not explained by subjects obtaining more extrinsic rewards following selection of *free* compared to *forced* options. Obtained reward proportions were not significantly different in the low (following selection of *free* vs. *forced*, 0.493 vs. 0.509, $\chi^2$ = 0.622, $p$ = 0.430) or medium (0.730 vs. 0.750, $\chi^2$ = 1.83, $p$ = 0.176) probability blocks. In contrast, in high probability blocks, subjects received significantly fewer rewards on average after *free* selection than after *forced* selection (0.989 vs. 1, $\chi^2$ = 9.97, $p$ = 0.002). In this block, reward was

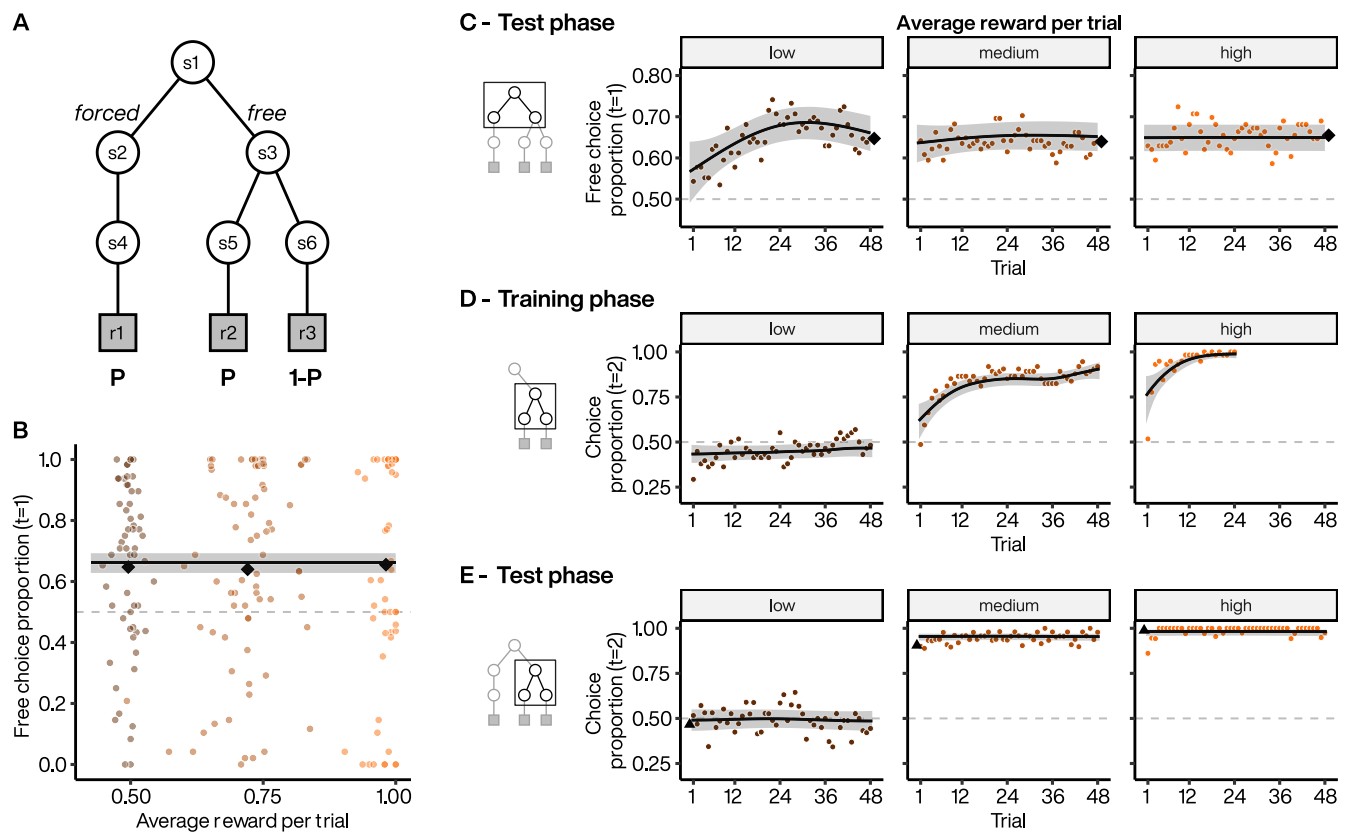

**Fig 2. Choice preference across different average extrinsic reward probabilities. A.** Experiment 1 task design where maximal extrinsic reward probabilities increased equally across *free* and *forced* options. **B.** Subject preference for *free* option during 1st-stage. Colored points indicate individual subject mean choice preference per block, plotted against the average obtained rewards. Black diamonds indicate the average of subject means per block. Line indicates the estimated choice preference from a GAMM, with 95% CI. **C.** Dynamics of *free* option preference across test phase blocks for low (left), medium (middle) and high (right) extrinsic reward probabilities. Each point represents the average *free* option preference as a function of trial within a block (smoothed with a 2-point moving average). Diamonds: as in B. Lines indicate the estimated choice preference from a GAMM, with 95% CI. **D** to **E.** Dynamics of the selection of the most rewarded 2nd-stage targets in *free* option for low (left), medium (middle) and high (right) blocks during the training (D) and test (E) phases. Note that in left panels, the extrinsic reward probability is equal for the two 2nd-stage targets (P = 0.5) and that in the right panel (P = 1), 24 trials were sufficient to train the subjects (see Materials and Methods). For P > 0.5, choice proportion indicates choice of the fractal associated with the higher reward probability. Triangles represent the final average selection at the end of the training phases. Lines: as in C.

fully deterministic, and *forced* selection always led to a reward, whereas *free* selections could lead to missed rewards if subjects chose the incorrect target. Choice preference in the deterministic condition cannot be explained by post-choice revaluation, which appears to occur only after choosing between closely valued options [30,34]. In this condition there is no cognitive dissonance to resolve when the choice is between a surely rewarded action and a never rewarded action.

## Trading extrinsic rewards for choice opportunities

Since manipulating the overall expected reward did not alter choice seeking behavior at the group-level, we investigated the effect of changing the relative expected reward between 1st-stage options. In experiment 2, we tested a new group of 36 subjects for whom we decreased the relative objective value of the *free* versus *forced* options. This allowed us to assess the point at which these options were equally valued and potentially reversed to favor the initially non-preferred (*forced*) option (Fig 3A). Thus, we assessed the value of choice opportunity by

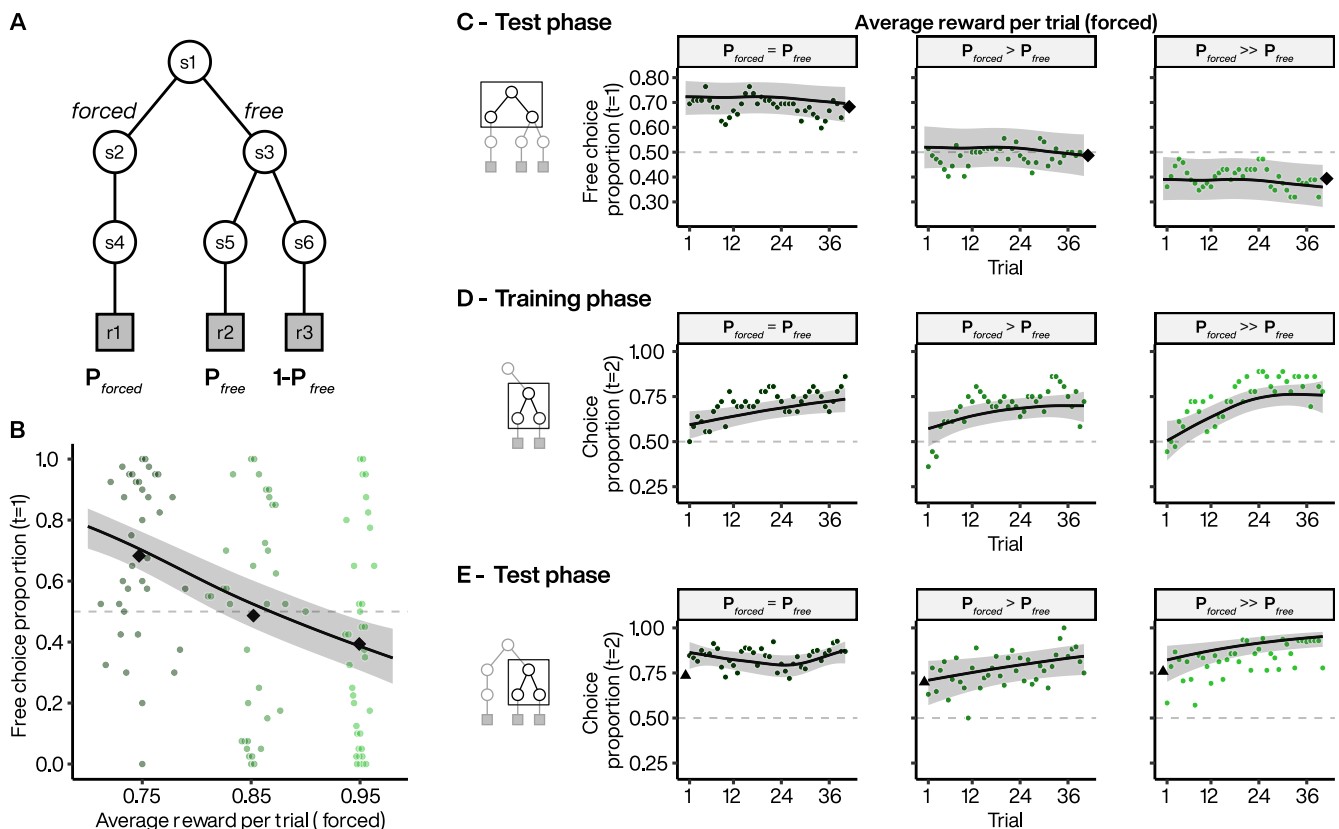

**Fig 3. Choice preference across different relative extrinsic reward probabilities. A.** Experiment 2 task design where extrinsic reward probably is always at P = 0.75 for the highly rewarded target in *free* options but varied from 0.75 to 0.95 across 3 blocks for *forced* options. **B.** Subject preference for *free* option during 1st-stage. Colored points indicate individual subject mean choice preference per block, plotted against the average rewards obtained in *forced* option. Black diamonds indicate the average of subject means per block. Line indicates the estimated choice preference from a GAMM, with 95% CI. **C.** Dynamics of *free* option preferences across test phase blocks when extrinsic reward probabilities of *forced* options were set at 0.75 (left), 0.85 (middle) and 0.95 (right). Each point represents the average *free* option preference as a function of trial within a block. Diamonds: as in B. Lines indicate the estimated choice preference from a GAMM, with 95% CI. **D to E.** Dynamics of the selection of the most rewarded 2nd-stage targets in *free* option when extrinsic reward probabilities of *forced* options are set at 0.75 (left), 0.85 (middle) and 0.95 (right) during the training (D) and test (E) phases. Triangles represent the final average selection at the end of the training phases. Lines: as in C.

increasing the reward probabilities following *forced* selection (block 1: $P_{forced}$ = 0.75; block 2: $P_{forced}$ = 0.85; block 3: $P_{forced}$ = 0.95), while keeping the reward probabilities following *free* selection fixed ($P_{free}$|a1 = 0.75, $P_{free}$|a2 = 0.25 for all blocks).

As in experiment 1, we found that subjects preferred choice when the extrinsic reward probabilities of the *free* and *forced* options were equal (block 1: 68% 1st-stage choice in favor of *free*; Fig 3B, dark green). Increasing the reward probability associated with the *forced* option significantly reduced choice preference ($\chi^2$ = 11.8, $p$ < 0.001, Fig 3B) to 49% (block 2) and 39% where the preference for *free* versus *forced* choice was reversed (block 3; see Fig C(A) in S1 Text for individual subject data). We estimated the population average preference reversal point at $P_{forced}$ = 0.88, indicating that indifference was obtained on average when the value of the *forced* option was 17% greater than that of the *free*. We found that subjects' preference remained constant across trials when reward probabilities were equal ($p$ > 0.1 for nonlinear smooth by trial; Fig 3C, left panel). Although reduced overall, the selection of the *free* option also did not vary significantly across trials in blocks 2 and 3 ($p$ > 0.1 for nonlinear smooths by trial, respectively). Furthermore, as in experiment 1, subjects acquired preference for the most

rewarded 2nd-stage targets during the learning phase (Fig 3D) and continued to express this preference during the test phase in all three blocks (Fig 3E). Thus, the decrease in choice preference was not related to failure to learn the reward contingencies during the training phase. Finally, RTs decreased as a function of $P_{forced}$ increased (estimated trend = -0.367, 95% CI = [-0.694, -0.024], $p$ = 0.020; Fig B(B) in S1 Text).

Although decreasing the relative value of the *free* option reduced choice preference, most subjects did not switch exclusively to the *forced* option. Even in block 3, where the *forced* option was set to be rewarded most frequently ($P_{forced}$ = 0.95 versus $P_{free}$ = 0.75), 32/36 subjects selected the *free* option in a non-zero proportion of trials. Since exclusive selection of the *forced* option would maximize extrinsic reward intake, continued *free* selection indicates a persistent appetency for choice opportunities despite their diminished relative extrinsic value.

We also asked subjects in experiment 2 to estimate the extrinsic reward probabilities associated with each 2nd-stage fractal image. They did so by placing a cursor on a visual scale from 0 to 100 after completing the test phase of each condition. We found that the subjects were relatively accurate at estimating the reward probabilities for fractals in both *free* and *forced* trials (Fig 4A), with mild underestimation (overestimation) at higher (lower) reward probabilities (estimated trend *forced* = 0.749, 95% CI = [0.683, 0.816], $t$ = 22.2, $p <$ 0.001; estimated trend *free* = 0.735, 95% CI = [0.640, 0.831], $t$ = 15.2, $p <$ 0.001), which did not differ significantly between these trial types (estimated trend difference = 0.014, 95% CI = [-0.102, 0.130], $t$ = 0.237, $p$ = 0.812). This suggests that preference for the *free* option was not due to differential distortion in estimating the frequency of rewards (Fig 4B). In addition, subjects did not overestimate the worst reward probability sufficiently to explain their preference. Note that by design such an overestimated probability must be greater than or equal to the best outcome reward probability in order to exceed or equal the expected value of the *forced* option. Say, for a true best outcome reward probability P = 0.75, the expected value for the *forced* option is 0.75 since the computer always selects the best target, and a subject would have to believe that the worst outcome probability in the *free* option is $\geq$ 0.75. This is not the case, and the average estimate for the worst rewarded fractal (mean = 0.35) is significantly below the 0.75 needed to match the expected value of the *forced* option ($t$ = -20.1, $p <$ 0.001). Finally, we also asked subjects to estimate the reward probability for the 2nd-stage fractals that were never chosen by the computer in the *forced* option (Fig 4A, putative programmed reward probabilities = 0.25, 0.15, 0.05). These probability estimations were not based on direct experience, and subjects appear to have inferred that it was 1-P from their experience in *free* trials, or alternatively this may be the consequence of a kind of counterfactual prior belief [35].

## Reinforcement-learning model of choice seeking

We next sought to explain individual variability in choice behavior using a value-based decision-making framework. We first used mixed logistic regression to examine whether rewards obtained from 2nd-stage actions influenced 1st-stage choices. We found that obtaining a reward on the previous trial significantly increased the odds that subjects repeated the 1st-stage selection that ultimately led to that reward (odds ratio rewarded/unrewarded on previous trial: 1.72, 95% CI = [1.46, 2.03], $p <$ 0.001). This suggest that subjects may continue to update their extrinsic reward expectations based on experience during the test phase. We therefore leveraged temporal-difference reinforcement learning (TDRL) models to characterize choice preference.

We fitted TDRL models to individual data using two distinct features to capture individual variability across different extrinsic reward contingencies. The first feature was a free choice bonus added to self-determined actions as an intrinsic reward. This can lead to overvaluation

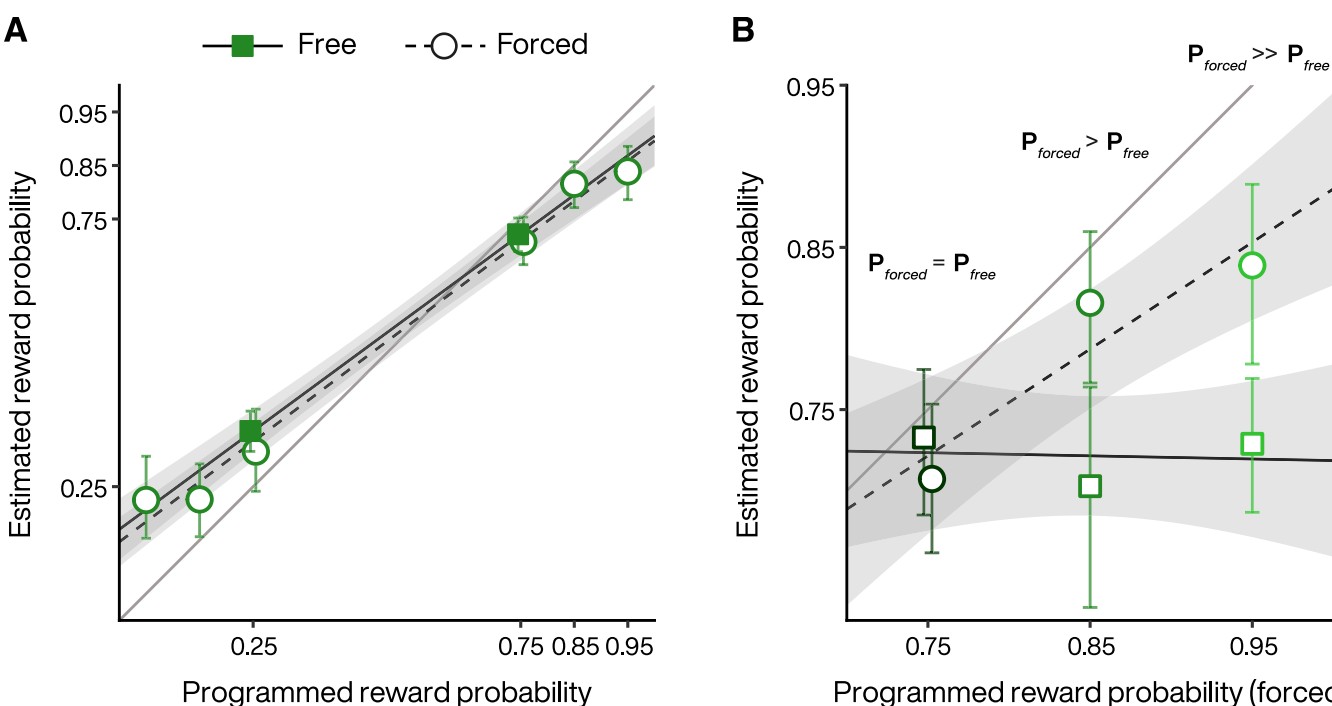

**Fig 4. Subject estimates of extrinsic reward probabilities in experiment 2. A.** Each circle (forced) or square (free) represents the average estimate of each 2nd-stage images experienced or viewed by the subjects across the three blocks. The estimates for fractals in the *free* option were averaged across the blocks but were similar both for the poorly rewarded targets (0.39, 0.35 and 0.32 for $P_{forced}$ = 0.75, 0.85 and 0.95 respectively) and the highly rewarded targets (0.73, 0.70 and 0.73 for $P_{forced}$ = 0.75, 0.85 and 0.95 respectively)). Black lines with shading indicate the estimated trends from a GLMM, with 95% CI. **B.** Same as in A but only for the highly rewarded targets (i.e., the most selected, see Fig 3). Note that in contrast to A., the abscissa in B. refers to the maximal reward probability for the *forced* option. $P_{free}$ did not vary significantly (estimated trend = -0.019, 95% CI = [-0.331, 0.293], p = 0.902), consistent with the maximal reward probability for the *free* option being constant at 0.75. Black lines with shading indicate the estimated trends from a GLMM, with 95% CI.

of the *free* option via standard TD learning. The second feature modifies the form of the future value estimate used in the TD value iteration, which in common TDRL variants is, or approximates, the best future action value (Q-learning or SARSA with softmax behavioral policy, respectively). We treated both Q-learning and SARSA together as optimistic algorithms since they are not highly discriminable with our data (Figs D-E in S1 Text). We compared this optimism with another TDRL variant that explicitly weights the best and worst future action values (Gaskett's $\beta$-pessimistic model [36]), which could capture avoidance of choice opportunities through increased weighting of the worst possible future outcome (pessimistic risk attitude). For example, risk is maximal in the high reward probability block in experiment 1 since selection of one 2nd-stage target led to a guaranteed reward (best possible outcome) whereas selection of the other target led to guaranteed non-reward (worst possible outcome).

We found that TDRL models captured choice preference across subjects and conditions (Fig 5A; see also Fig E in S1 Text). For 80% (33/41) of subjects in experiment 1 who preferred the *free* option on average (>50% across all trials), the best model incorporated overvaluation of rewards obtained from *free* actions (Fig 5B, Table A in S1 Text). Therefore, optimistic or pessimistic targets alone were insufficient to explain individual choice preference across different extrinsic reward contingencies. Since some subjects preferred the *forced* option for one or more experimental conditions (Fig A in S1 Text), we examined whether individual parameters from the fitted TDRL models were associated with choice preference. We did not find significant correlations of average choice preference with that 2nd-stage learning rates ($r$ = 0.140, $t$ = 1.02, $p$ = 0.311), softmax inverse temperatures ($r$ = 0.033, $t$ = 0.245, $p$ = 0.807) or tendencies

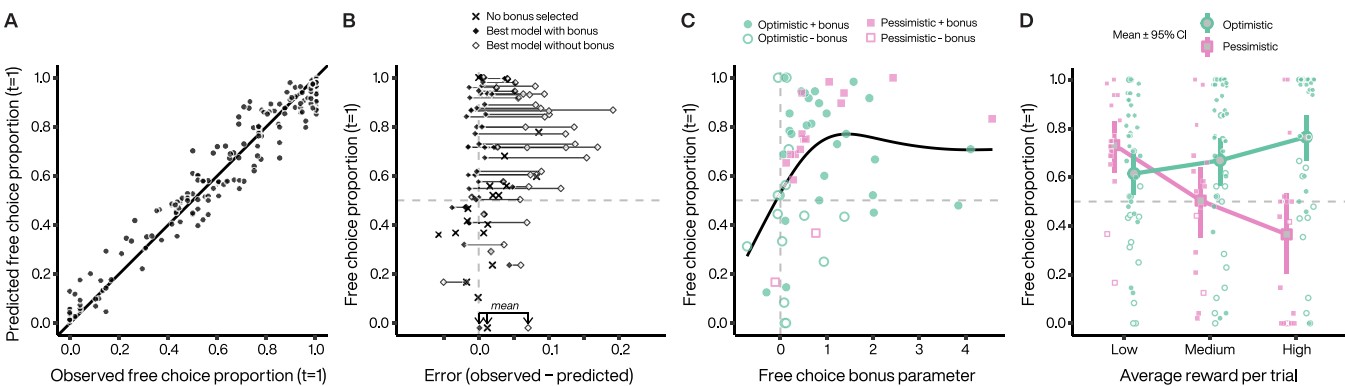

**Fig 5. Reinforcement learning models capture individual choice preference. A.** Free choice proportions predicted by the winning model plotted against observed free choice proportions for each condition for each subject in experiment 1. **B.** Obtained free choice proportion as a function of model error, averaged over all conditions. For subjects where the selected model did not include a free choice bonus, only one symbol (X) is plotted. For subjects where the best model included a free choice bonus, two symbols are plotted and connected by a line. Filled symbols represents the fit error with the selected model, and the open symbols represents the next best model that did not include a free choice bonus. **C.** Bonus coefficients increase as a function of subjects' preference for *free* options irrespectively of the target policy they used when performing the task. Choice preference from low probability blocks (P = 0.5). Filled symbols indicate that the best model included a free choice bonus. Open symbols indicate that the best model did not include a free choice bonus, and the bonus value plotted is taken from the best model fit with an added free choice bonus. Line illustrates a generalized additive model smooth. **D.** Pessimistic subjects significantly decreased their *free* option preference as a function of extrinsic reward probabilities (estimated trend = -4.58, 95% CI = [-7.45, -1.71], *p* = 0.002). This decrease was significantly different from optimistic subjects (*z* = -4.81, *p* < 0.001), who increased their choice preference (estimated trend = 3.76, 95% CI = [1.94, 5.57], p < 0.001). Symbol legend from C applies to the small points representing individual means in D. Error bars represent 95% CI from bootstrapped individual means.

to repeat 1st-stage choices (*r* = 0.200, *t* = 1.54, *p* = 0.128). We did find that the magnitude of the free choice bonus was significantly associated with increasing choice preference (*r* = 0.370, *t* = 2.94, *p* = 0.005, Fig 5C). We also found a significant correlation with the relative weighting between the worst and best possible outcomes in the *β*-pessimistic target (*r* = 0.340, *t* = 2.69, *p* = 0.009), with increasing weighting of the best outcome (*β*→1, see Material and Methods) being associated with increasing average choice preference. The pessimistic target best fitted about 28% (16 of 58) of the subjects in experiment 1, and most pessimistic subjects (14 of 16) were best fitted with a model including a free choice bonus to balance risk and decision attitudes across reward contingencies. In experiment 1, we introduced risk by varying the difference in extrinsic reward probability for the best and worst outcome following *free* selection. The majority of so-called 'pessimistic subjects' preferred choice when extrinsic reward probabilities were low, but their weighting of the worst possible outcome significantly decreased this preference as risk increased (Fig 5D, pink). Thus, the most pessimistic subjects avoided the *free* option despite rarely or never selecting the more poorly rewarded 2nd-stage target during the test phase.

We also fitted the TDRL variants to individual data from experiment 2 and found that a free choice bonus was also necessary to explain choice preference across extrinsic reward contingencies in that experiment. Five subjects (of 36) were best fitted using the *β*-pessimistic target (see Fig F in S1 Text) although this may be a conservative estimate since we did not vary risk in experiment 2.

An alternative model for choice preference is a 1st-stage choice bias. While a bias towards the *free* option can generate choice preference, a key difference is that a free choice bonus can lead to altered action-value estimates since the inflated reward enters into subsequent value updates to modify Q-values. A bias, on the other hand, can generate free choice preference but will not enter directly into subsequent value updates (hence it will not directly affect action-value estimates). Our initial decision to implement a free choice outcome bonus was motivated

by prior experiments showing that cues associated with reward contingencies learned in a *free* choice context were overvalued compared to cues associated with the same reward contingencies learned in a *forced* choice context [5,37]. We found that free choice outcome bonus fits our data better than a softmax bias (Fig G in S1 Text), suggesting that most subjects may update action-values following *free* and *forced* selections rather than applying a static bias.

## Influence of action-outcome coherence on choice seeking

We next asked whether choice preference was related to personal control beliefs. To do so, we manipulated the coherence between an action and its consequence over the environment. In experiment 3, we tested the relationship between preference for choice opportunity and the physical coherence of the terminal action by directly manipulating the controllability of $2^{nd}$-stage actions. We modified the two-stage task to introduce a mismatch between the subject's selection of the $2^{nd}$-stage target and the target ultimately displayed on the screen by the computer (Fig 6A and Materials and Methods section). We did this by manipulating the probability that a $2^{nd}$-stage target selected by a subject would be swapped for the $2^{nd}$-stage target that had not been selected. That is, on coherent trials, a subject selecting the fractal on the right side of the screen would receive visual feedback indicating that the right target had been selected. On incoherent trials, a subject selecting the fractal on the right side would receive feedback that the opposite fractal target had been selected (i.e., the left target).

To ensure that all other factors were equalized between the two $1^{st}$-stage choices, we implemented target swaps following both *free* and *forced* selections by adding an additional state to our task (Fig 6A). In one block of trials, the incoherence was set to 0 and every subject action in the $2^{nd}$-stage led to a coherent selection of the second target. In the other blocks, we set incoherence to 0.15 or 0.3, resulting in lower controllability between target choice and target selection (e.g., 85% of the time, pressing the left key will select the left target, and in 15% the right target). We set all the extrinsic reward probabilities associated with the different fractal targets to P = 0.75. Since all $2^{nd}$-stage actions had the same expected value, the experiment was objectively uncontrollable because the probability of reward was independent of all actions [18]. Moreover, equal reward probabilities ensured that outcome diversity [38,39], outcome entropy [40], and instrumental divergence [41] did not contribute to choice preference since these were all equal between the *forced* and *free* options.

The same group of participants who performed experiment 2 also performed experiment 3 (*n* = 36). We compared the two similar conditions (block 1 from experiment 3 with full coherence and block 1 from experiment 2 with equal extrinsic rewards), which differed in that choosing the *forced* option resulted in the obligatory selection of the same fractal (experiment 2) or one of two fractals randomly selected by the computer (experiment 3). We found that choice preference was similarly high (mean for experiment 3 = 69%, bootstrapped 95% CI = [58, 78], and for experiment 2 = 68%, 95% CI = [59, 76]) and did not differ significantly (mean difference (experiment 2 –experiment 3) = -0.013, bootstrapped 95% CI = [-0.106, 0.082], *p* = 0.792), indicating that subjects' choice preference in experiment 2 was not related to lack of action variability following *forced* selection per se. Moreover, we found that choice preference in these two blocks was significantly correlated (Spearman's *r* = 0.538, bootstrapped 95% CI = [0.186, 0.779], *p* = 0.003), highlighting a within-subject consistency in choice preference.

Increasing incoherence decreased $1^{st}$-stage choice preference by 2.4 and 5.4 points in blocks 2 and 3 respectively compared to block 1 (full controllability; Fig 6B). This decrease was not significant (estimated trend = -1.55, 95% CI = [-4.45, 1.36], *p* = 0.298; see Fig C in S1 Text for individual subjects). Consistent with experiments 1 and 2, choice preference was expressed immediately after the training phase and remained constant throughout the different blocks

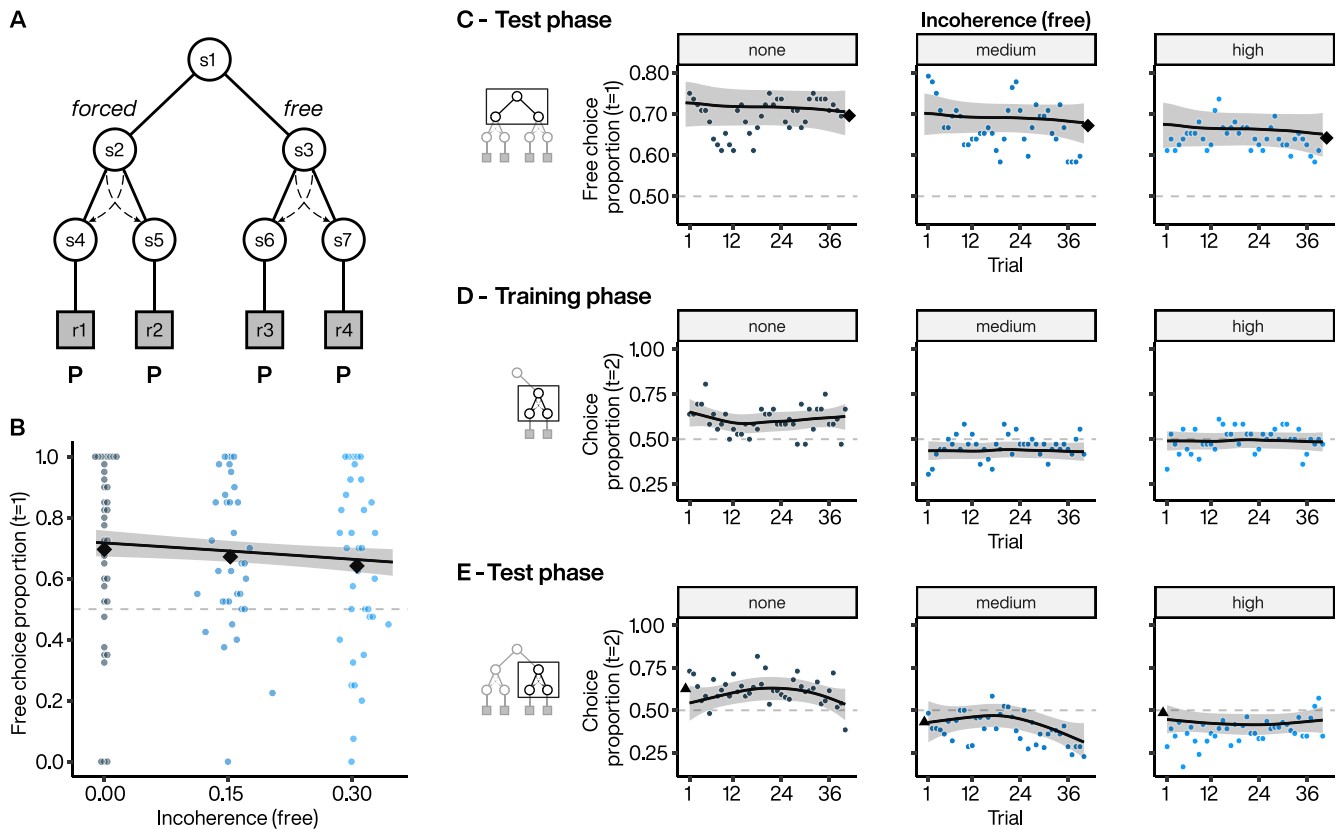

**Fig 6. Choice proportion across different levels of action-outcome incoherence. A**. Experiment 3 task design using seven-state task design where we manipulated the incoherence. We added an additional state following the *forced* option in order to manipulate the incoherence in both *free* and *forced* options. Dashed arrows represent potential target swaps on incoherent trials, the probability of which varied from 0 to 0.3 across different blocks. At incoherence = 0, the 2nd-stage target presented to the subject matched their selected target. In this task version an initial light red V-shape was associated with the target initially selected by the subject and was then changed to darker red either above the same target (i.e., overlap on coherent trials) or above the other target ultimately selected by the computer (i.e., incoherent trials). Extrinsic reward probabilities for all the 2nd-stage targets were set at P = 0.75. **B**. Subject preference for *free* option during 1st-stage. Colored points indicate individual subject mean choice preference per block, plotted against the incoherence level. Black diamonds indicate the average of subject means per block. Line indicates the estimated choice preference from a GAMM, with 95% CI. **C**. Dynamics of *free* option preference across test phase blocks for incoherence set at 0 (i.e., none, left), 0.15 (i.e., medium, middle) and 0.30 (i.e., high, right). Each point represents the average *free* option preference as a function of trial within a block. Diamonds: as in B. Lines indicate the estimated choice preference from a GAMM, with 95% CI. **D, E**. Dynamics of the selection of the two 2nd-stage targets (equally rewarded) in *free* options across the blocks for incoherence levels set at 0 (left), 0.15 (middle) and 0.30 (right) during the training (D) and test (E) phases. Triangles represents the final average selection at the end of the training phases. Lines: as in C.

(Fig 6C–6E). Increasing incoherence speeded 1st-stage RTs for *free* compared to *forced* selection (median difference *free–forced* = -0.060, 95% CI = [-0.110, -0.016], $p = 0.009$), with an additive effect of faster 1st-stage RTs with increasing incoherence (estimated trend = -0.279, 95% CI = [-0.570, 0], $p = 0.040$, Fig B(C) in S1 Text).

We found that choice preference depended on the choice made on the previous trial (Fig 7A), with a significant main effect of previous 1st-stage choice ($\chi^2 = 451$, $p < 0.001$) as well as a significant interaction between the previous 1st-stage choice (*free* or *forced*) and the degree of incoherence ($\chi^2 = 11.5$, $p < 0.001$). Specifically, the frequency of *free* selections after a *forced* selection decreased significantly with incoherence (estimated trend = -2.68, 95% CI = [-4.92, -0.435], $z = -2.34$, $p = 0.019$), whereas the frequency of repeating a previous *free* selection did not change significantly with incoherence (estimated slope = -0.078, 95% CI = [-2.23, 2.07], $z = -0.071$, $p = 0.943$). Thus, as incoherence increased, subjects tended to stay more with the *forced* option, while maintaining a preference to repeat *free* selections.

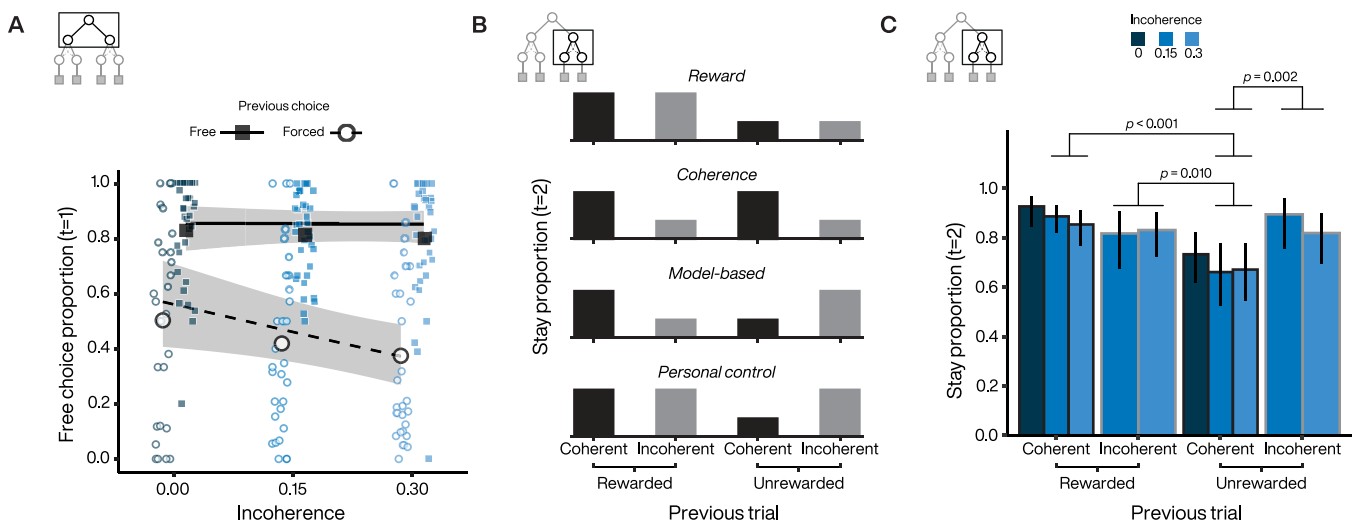

**Fig 7. Controllability alters sequential decisions at 1st and 2nd-stages. A.** First-stage free choice proportion after a *free* and *forced* trial as a function of incoherence. **B.** Second-stage stay probabilities for the different action-state-reward trial types. Each subpanel represents a putative strategy followed by the subject. **C.** Estimated 2nd-stage stay probabilities. Error bars represent 95% CI. P-values are displayed for significant pairwise comparisons and adjusted for multiple comparisons. Statistics for significant comparisons are: Unrewarded&Coherent vs. Rewarded&Coherent (log odd ratio (OR) = -1.20, $z$ = -5.4, $p < 0.001$), Unrewarded&Coherent vs. Unrewarded&Incoherent (log OR = -1.11, $z$ = -3.62, $p$ = 0.002), Unrewarded&Coherent vs. Rewarded&Incoherent (log OR = -0.843, $z$ = -3.12, $p$ = 0.010). Statistics for non-significant comparisons are: Rewarded&Coherent vs. Unrewarded&Incoherent (log OR = 0.083, $z$ = 0.260, $p$ = 0.994), Rewarded&Coherent vs. Rewarded&Incoherent (log OR = 0.354, $z$ = 1.70, $p$ = 0.323), Unrewarded&Incoherent vs. Rewarded&Incoherent (log OR = 0.271, $z$ = 0.760, $p$ = 0.872).

The sustained repetition of *free* selections across the different levels of incoherence suggests that subjects may have been acting as if to maintain control over the task through self-determined 2nd-stage choices. Although the task was objectively uncontrollable since all terminal action-target sequences were rewarded with the same probability, subjects may have developed structure beliefs based on local reward history and target swaps, which could be reflected in 2nd-stage patterns of choice. Thus, subjects may have followed a strategy based on reward feedback by repeating only actions associated with a previous reward (illusory maximization of reward intake; Fig 7B, first panel). Alternatively, they could have followed a strategy based on action-outcome incoherence feedback and thus avoided trials associated with a previous target swap (illusory minimization of incoherent states; Fig 7B, second panel). However, subjects may have also employed another classic strategy known as "model-based" where agents use their (here illusory) understanding of the task structure built from all the information provided by the environment (Fig 7B, third panel) [42]. Under this strategy, subjects try to integrate both the reward and target-swap feedback to select the next target in order to maximize reward. For example, an incoherent but rewarded trial would lead to a behavioral switch if the subject has integrated the information provided by the environment (i.e., the target swap induced by the computer), signaling that the other target is actually rewarded (see second bar on third panel of Fig 7B). Finally, an alternative strategy could rely on maintaining control in situations where control is threatened by adverse events (e.g., action-outcome incoherence). Subjects would thus ignore any choice outcome from trials with reduced controllability (incoherent trials) and repeat their initial choice selection (Fig 7B, fourth panel).

We found a significant interaction between last reward and last target swap ($\chi^2$ = 10.9, $p$ = 0.001). Results of the stay behavior during 2nd-stage choice following *free* selection suggests that subjects consider the source of reward when choosing between the different fractal targets (Fig 7C). Indeed, when their action was consistent with the state they were choosing (i.e., the coherent fractal target feedback), they took the reward outcome into account to adjust their

behavior on the next trial, either by staying on the same target when the trial was rewarded or by switching more to the other one when no reward was delivered. However, subjects were insensitive to outcomes from incoherent trials as they maintained the same strategy (staying) during subsequent trials, regardless of whether they were previously rewarded or not. We suggest this strategy reflects an attempt to maintain a consistent level of control over the environment at the expense of the task goal of maximizing reward intake. Note that 2nd-stage reaction times were not significantly modulated by the presence or absence of a target swap, suggesting that subjects were not confused following incoherent trials (see Fig H in S1 Text).

## Discussion

Animals prefer situations that offer more choice to those that offer less. Although this behavior can be reliably measured using the two-stage task design popularized by Voss and Homzie [8], their conclusion that choice has intrinsic value is open to debate. To rule out alternative explanations for choice-seeking, we performed three experiments in which we clearly separated learning of reward contingencies from testing of choice preference. Our experiments indicate that a sustained preference for choice opportunities associated with subjects' decision attitudes (overvaluation of reward outcome) can be modulated by extrinsic reward properties including reward probability and risk as well as by controllability of the environment.

In the first and second experiments, we varied the reward probabilities associated with terminal actions following *free* and *forced* selection. Consistent with previous studies [2,4], subjects preferred the opportunity to make a choice when expected rewards were equal between terminal actions (P = 0.5). Surprisingly, subjects also preferred choice when we increased the value difference between terminal actions in the *free* option, while keeping the *maximum* expected reward equal in the *free* and *forced* options (P > 0.5). This sustained preference for choice is potentially economically suboptimal since making a free choice carries the risk of making an error leading to lowered reward intake. The persistence of this preference for free choice even when reward delivery was deterministic (P = 1), makes it unlikely that this preference was due to an underestimation of *forced* trials due to poor learning of reward contingencies.

Subjects appeared to have understood the reward contingencies as evidenced by their consistent preference for the highest-rewarded 2nd-stage fractal, which was acquired during the training phase and expressed during the test phase. This stable 2nd-stage fractal selection, together with the immediate expression and maintenance of 1st-stage choice preference, renders unlikely accounts based on curiosity, exploration or variety seeking since varying the probability of rewards did not decrease choice preference for about two third of the subjects (i.e., optimistic subjects). Regarding variety seeking as an alternative, we can distinguish potential from expressed variety. Subjects in the deterministic condition (P = 1) of experiment 1 did not seek to express variety since most exclusively selected the best rewarded 2nd-stage target during the test phase. However, subjects may prefer the *free* option in this condition because there is still potential variety that they rarely or never act on. Thus, in experiments 1 and 2, two different computational goals (maximizing potential variety versus maximizing control) can be achieved using the same algorithm (overvaluating *free* outcomes). However, there is a feature of our experiment 3 that argues in favor of maximizing control over potential variety per se. In experiment 3 we modified the state structure of the task so that choosing the *forced* option led to the computer randomly selecting a fractal target in the 2nd-stage with equal probability. Insofar as potential variety can be equated with the variability of possible actions, then potential variety (and ultimately expressed variety) is maximal for the *forced* option. Yet subjects continue to prefer the *free* option, suggesting that, despite more variety following the

*forced* option, subjects preferred to control 2^nd^-stage selections. That is, subjects preferred to "keep their options open" rather than simply "keeping all options open".

Selection-based accounts also have trouble explaining the pattern of results we observed. The idea that post-choice revaluation specifically inflates expected outcomes after choosing the *free* option can explain choice-seeking when all terminal reward probabilities are equal. However, post-choice revaluation cannot explain choice preference when the terminal reward probabilities in the *free* option clearly differ from one another, since revaluation appears to occur only after choosing between closely valued options [30,34]. That is, there is no cognitive dissonance to resolve when reward contingencies are easy to discriminate, and no preference for choice should be observed when the choice is between a surely (i.e., deterministically) rewarded action and a never rewarded action. The existence of choice preference in the deterministic condition (P = 1) also cannot be explained by an optimistic algorithm such as Q-learning, since the maximum action value is equal to the maximum expected value, and the value of the *free* option is not biased upwards under repeated sampling [33].

Although standard Q-learning could not capture variability across different terminal reward probabilities, we found that combining two novel modifications to TDRL models was able to do so. The first feature was a free choice bonus—a fixed value added to all extrinsic rewards obtained through free actions—that can lead to overvaluation of the free option via standard TD learning. This bonus implements Beattie and colleagues' concept of *decision attitude*, the desire to make or avoid decisions independent of the outcomes [12]. The second feature modifies the form of the future value estimate in the TD value iteration. Zorowitz and colleagues [33] showed that replacing the future value estimate in Q-learning with a weighted mixture of the best and worst future action values [36] can generate behavior ranging from aversion to preference for choice. The mixing coefficient determines how optimism (maximum of future action values, total risk indifference) is tempered by pessimism (minimum of future action values, total risk aversion). In experiment 1, we found that 28% of subjects were best fitted with a model incorporating pessimism, which captured a downturn in choice preference with increasing relative value difference between the terminal actions in the *free* option. Importantly however, individual variability in the TD future value estimates alone did not explain the pattern of choice preference across target reward probabilities, and a free choice bonus was still necessary for most subjects. Thus, the combination of both a free choice bonus (decision attitude) and pessimism (risk attitude) was key for explaining why some individuals shift from seeking to avoiding choice. This was unexpected because the average choice preference in experiment 1 was not significantly different across reward manipulations, highlighting the importance of examining behavior at the individual level. Here, we examined risk using the difference between the best and worst outcomes as well as relative value using probability (see [43]). It may be the case that variability is also observed in how individuals balance the intrinsic rewards with other extrinsic reward properties that can influence choice preference, such as reward magnitude [43].

We provided monetary compensation for subject participation, and they were instructed to earn as many rewards as possible. We further motivated subjects with extra compensation for performing the task correctly (see Materials and methods). One limitation of our experiments is that subjects were not compensated as a direct function of the reward feedback earned in the task (i.e., each reward symbol did not lead to fixed monetary amount that was accumulated and paid out exactly at the end of the session). It is plausible that direct monetary feedback could lead to differences at the group and individual level. For example, increasing the direct monetary payoff associated with every successful action could be expected to eventually abolish choice preference for most if not all subjects (in the deterministic condition in experiment 1 a subject risks losing a potentially large payoff due to choosing incorrectly at the 2^nd^-stage).

Understanding how direct reward feedback influences choice preference will be important for better understanding the role of intrinsic motivation for choice in real world decision making.

Prior studies have shown that different learning rates for positive and negative prediction errors can account for overvaluation of cues learned in free choice versus forced selection contexts (e.g. [37]). Our decision to use a scalar bonus was motivated by the results from Chambon and colleagues, who showed that a model that directly altered the free choice outcomes was able to replicate the results of different positive and negative learning rates. This indicates that these mechanisms mimic each other and are difficult to distinguish in practice, and that specifically designed tasks are probably necessary for definitive claims about when one or the other of these mechanisms can be ruled out. Also, to produce choice preference, differential learning rates would have to be fit for both the *free* and *forced* options, which would greatly increase the number of parameters in our task (to handle 1$^{st}$ and 2$^{nd}$ stages). Finally, the free choice bonus is conceptually easier to apply for understanding prior studies where choice preference is demonstrated where there is no obvious learning (e.g., one-shot questionnaires about fictitious outcomes [3]).

Why are choice opportunities highly valued? It may be that choice opportunities have acquired intrinsic value because they are particularly advantageous in the context of the natural environment in which the learning system has evolved. Thus, choice opportunities might be intrinsically rewarding because they promote the search for states that minimize uncertainty and variability, which could be used by an agent to improve their control over the environment and increase extrinsic reward intake in the long run [44,45]. Developments in reinforcement learning and robotics support the idea that both extrinsic and intrinsic rewards are important for maximizing an agent's survival [46–48]. Building intrinsic motivation into RL agents can promote the search for uncertain states and facilitate the acquisition of skills that generalize better across different environments, an essential feature for maximizing an agent's ability to survive and reproduce over its lifetime, i.e., its evolutionary fitness [46].

The intrinsic reward of choice may be a specific instance of more general motivational constructs such as autonomy [15,16], personal causation [19], effectance [20], learned helplessness [49], perceived behavioral control [21] or self-efficacy [17], which are key for motivating behaviors that are not easily explained as satisfying basic drives such as hunger, thirst, sex, or pain avoidance [22]. Common across these theoretical constructs is that control is intrinsically motivating only when the potential exists for agents to determine their own behavior, which when realized can give rise to a sense of agency and, in turn, strengthens the belief in the ability to exercise control over one's life [50]. Thus, individuals with an *internal* locus of control tend to believe that they, as opposed to external factors such as chance or other agents, control the events that affect their lives. Crucially, the notion of locus of control makes specific predictions about the relationship between preference for choice—choice being an opportunity to exercise control—and the environment: individuals should express a weaker preference for choice when the environment is adverse or unpredictable [51]. This prediction is consistent with what is known about the influence of environmental adversity on control externalization: individuals exposed to greater environmental instabilities tend to believe that external and uncontrollable forces are the primary causes of events that affect their lives, as opposed to themselves [52]. In other words, one would expect belief in one's ability to control events, and thus preference for choice, to decline as the environment becomes increasingly unpredictable.

In our third experiment, we sought to test whether it was specifically a belief in personal control over outcomes that motivated subjects by altering the controllability of the task. To do so, we introduced a novel change to the two-stage task where in a fraction of trials subjects experienced random swapping of the terminal states (fractals). Thus, subjects were subjected to trials where the terminal state was incoherent with their choice, and thus experienced

alterations in their ability to predict the state of the environment following their action. Incoherence occurred with equal probability following free and forced actions to equate for any value associated with swapping itself.

Manipulating the coherence at the 2nd-stage did not decrease the overall preference for free choice. However, we found a significant reduction in the propensity to switch from forced to free choice following action-target incoherence, suggesting that reducing the controllability of the task causes free choice to lose its attractiveness. This reduction in choice preference following incoherent trials is reminiscent of a form of locus externalization and is consistent with the notion that choice preference is driven by a belief in one's personal control. In this experiment, we focused on the value of control, and therefore equated other decision variables such as outcome diversity [38,39], outcome entropy [40], and instrumental divergence [41,53]. Further experiments are needed to understand how these variables interact with personal control in the acquisition of potential control over the environment.

Interestingly, when subjects selected the *free* option, the subsequent choice was sensitive to the past reward when the terminal state (the selected target) was coherent and the reward could therefore be attributed to the subject's action. In contrast, subjects' choices were insensitive to the previous outcome when the terminal state was incoherent and thus unlikely to be attributable to their choice. In other words, subjects appeared to ignore information about action-state-reward contingencies that was externally derived, and instead appeared to double down by repeating their past choice as if to maintain some control over the task. This behavior is consistent with observations suggesting that when individuals experience situations that threaten or reduce their control, they implement compensatory strategies to restore their control to its baseline level [54,55]. A variety of compensatory strategies for reduced control has been documented in humans. Thus, individuals experiencing loss of control are more likely to use placebo buttons—that is, to repeatedly press buttons that have no real effect—or to see images in noise or to perceive correlations when there are none [53]. We suggest that discounting of choice outcomes in incoherent trials, where personal control is threatened by unpredictable target swaps, is an occurrence of such a compensatory strategy. When faced with uncontrollability, participants would ignore any choice outcome and repeat their initial choice to compensate for the reduction in control, consistent with the broader psychology of control maintenance and threat-compensation literature (see 52 for a review).

Repetition of the initial choice following incoherent trials is more than mere perseveration (i.e., context-independent repetition) because it occurs specifically when the subject performs a voluntary (free) choice and witnesses an incoherence between that choice and its outcome. In this sense, it is reminiscent of the "try-and-try again" effect already observed in similar reinforcement-learning studies measuring individuals' sense of agency in sequences alternating free and forced choice trials [56] or the "choice streaks" exhibited by participants when making decisions under uncertainty [35]. Thus, when the situation is uncertain, or the feedback from their action is questionable or contradicts what they believe, participants try again and repeat the same choice until they have gathered enough evidence to make an informed decision, and change or not change their mind [57]. Similarly, the repetition of choices in Experiment 3 suggests that participants are reluctant to update their belief about their personal control and repeat the same action to test it again.

Finally, it should be noted that this strategy of ignoring choice outcomes—positive or negative—from uncontrollable contexts is at odds with a pure model-based strategy [42], where an agent could exploit information about action-state-reward contingencies whether it derived from their own choices (controllable context) or from the environment or another agent (uncontrollable context). Rather, it is consistent with work showing that choice seeking could emerge when self-determined actions amplify subsequent positive reward prediction errors

[5,37], and more generally with the notion that events are processed differently depending on individuals' beliefs about their own control abilities. Thus, positive events are amplified only when they are believed to be within one's personal control, as opposed to when they are not [37], or when they come from an uncontrollable environment [58]. A simple strategy consistent with the behavior we observed is one where subjects track rewards from self-determined actions but ignore rewards following target swaps that are incoherent with their actions. Alternatively, subjects may apply a modified model-based strategy where they attribute all positive outcomes to their credit, although here the high repetition rate would be due to illusory self-attribution on rewarded incoherent actions and model-based non-attribution on unrewarded incoherent actions. Key to both strategies, however, is that subjects modify their behavior based on recognizing when their self-determined actions produce incoherent outcomes.

Together, our results suggest that choice seeking, although influenced by individuals' attitudes in contexts where choosing involves risk, represents one critical facet of intrinsic motivation that may be associated with the desire for control. They also suggest that the preference for choice can compete with maximization of extrinsic reward provided by externally driven actions. Indeed, subjects favor positive outcomes associated with self-determined actions even if overall reward rate is lower than that from instructed actions. In general, the perception of being in control could then account for several aspects of our daily lives such as enjoyment during games [59] or motivation to perform demanding tasks [60]. Since our results showed inter-individual differences, it would be nonetheless important in the future to phenotype subject behaviors during choice-making to investigate how these individual traits can explain attitude differences when facing decisions and their consequences, as exemplified by the variety of attribution and explanation styles of individuals in the general population [61,62].

## Materials and methods

### Ethics statement

The local ethics committee (Comité d'Evaluation Éthique de l'Inserm) approved the study (CEEI/IRB00003888). Participants gave written informed consent during inclusion in the study, which was carried out in accordance with the declaration of Helsinki (1964; revised 2013).

### Participants

Ninety-four healthy individuals (mean age = 30 ±SD 7.32 years, 64 females) responded to posted advertisements and were recruited to participate in this study. Relevant inclusion criteria for all participants were being fluent in French, not treated for neuropsychiatric disorders, having no color vision deficiency and being aged between 18 and 45 years old. Out of these 94 subjects, 58 participated to experiment 1 and 36 to experiments 2–3. We gave subjects 40 euros for participating but they were informed that their basic compensation would be 30 euros and that an extra compensation of 10 euros would be added if they performed all trials correctly. We also asked subjects whether they were motivated by this extra-compensation and less than 12% reported not being motivated by it (11 out of 93 subjects who answered the question). The sample size was chosen based on previous reward-guided decision-making studies using similar two-armed bandit tasks and probabilistic choices [37,63,64].

### General procedure

The paradigm was written in Matlab using the Psychophysics Toolbox extensions [65,66]. It was presented on a 24 inch screen (1920 x 1080 pixels, aspect ratio 16:9). Subjects seat ~57 cm

from the center of the monitor. Our behavioral task design was designed as a value-based decision paradigm. All participants received written and oral instructions. They were told that the goal of the task was to gain the maximum number of rewards (a large green euro). They were informed about the differences between the different trial types and that the extrinsic reward contingencies experienced during the training phases remained identical during the test phases. After instructions, participants received a pre-training session of a dozen trials (pre-train and pre-test phases) in order to familiarize them with the task design and the keys they would have to press.

In our experiments, subjects performed repeated trials with a two-stage structure. In the 1st-stage they made an initial decision about what could occur in the 2nd-stage. Selecting the *free* option led to a subsequent opportunity to choose and selecting the *forced* option led to an obligatory computer-selected action. In the 2nd-stage, we presented subjects with two fractal images, one of them being more rewarded following *free* selection in experiment 1 (except for P = 0.5) and experiment 2. In experiments 1 and 2, the computer always selected the same fractal target following *forced* selection. In experiment 3 all fractal targets were equally rewarded and the computer randomly selected one of the two fractal targets following *forced* selection (50%). Following *forced* selection, the target to select with a key press was indicated by a grey V-shape above the target. Pressing the other key on this trial type did nothing and the computer waited for the correct key press to proceed further in the trial sequence. Either at the 1st- or 2nd-stage, after the subject's selection of the target, a red V-shape appears immediately after above the target to indicate the one they had selected (in experiment 3 blocks this red V-shape remains 250ms on the screen and eventually jumped with the target, see below).

## Experimental conditions

In experiment 1, fifty-eight subjects performed trials where the maximal reward probabilities were matched following *free* and *forced* selection. We varied the overall expected value across different blocks of trials, each of them being associated to different programmed extrinsic reward probabilities (P). Forty-eight subjects performed a version with 3 blocks (experiment 1a) with different extrinsic reward probabilities ranging from 0.5 to 1 (block 1: $P_{forced} = P_{free} = 0.5$; block 2: $P_{forced} = 0.75$, $P_{free}|a1 = 0.75$, $P_{free}|a2 = 0.25$; block 3: $P_{forced} = 1$, $P_{free}|a1 = 1$, $P_{free}|a2 = 0$; where a1 and a2 represent the two possible key presses associated with the fractal targets). Ten additional subjects performed the same task with 4 different blocks (experiment 1b) with extrinsic reward probabilities also ranging from 0.5 to 1 (P = 0.5 or 0.67 or 0.83 or 1 from block 1 to 4 respectively.) We used a GLMM to compare subjects who experienced four blocks (mean choice preference = 66%) to those that experienced three blocks (mean choice preference = 64%). There was no main effect for group ($\chi^2 = 0.093$, $p > 0.05$), nor a significant interaction with reward probability ($\chi^2 = 1.28$, $p > 0.05$). Therefore, we pooled data from these groups for analyses.

Experiment 2 was like experiment 1 (six states) except programmed extrinsic reward associated with the *forced* option were higher than the *free* option in two out of three blocks ($P_{forced} = 0.75$, 0.85 or 0.95). Reward probabilities following *free* selection did not change across the three blocks ($P_{free}|a1 = 0.75$, $P_{free}|a2 = 0.25$)

Experiment 3 consisted of a 7-state version of the two-stage task. Here, we manipulated the coherence between the subject selection of a 2nd-stage (fractal) target and the target ultimately displayed on the screen by the computer. Irrespectively of the target finally selected by the computer or the subjects, the extrinsic reward probability associated to all the 2nd-stage targets in *free* and *forced* trials was set at P = 0.75. Importantly, adding the 7th state in this last task version allowed the computer to swap the fractal 2nd-stage targets following both *free* and *forced*

selection. Thus, subjects did not perceive the weak coherence as a feature specific to the *free* condition. To prevent incoherent trials to be perceived as an informatic malfunction, preceding experiment 3, subjects received the written information that their computer have been extensively tested to detect any malfunction during the task and that therefore it was perfectly functional. We also provide visual feedback to the subjects after their key press to inform them that they did correctly select one of the keys (e.g., the right keys) but the computer ultimately swapped the final target displayed on the screen (e.g., the left target). This was to ensure that target swaps would not be perceived as their own errors of selection.

We associated unique fractal targets with each action in the 2nd-stage, and a new set was used for each block in all experiments. Colors of the 1st-stage targets were different between experiments. Positive or negative reward feedback, as well as the side of the 1st-stage and 2nd-stage target positions, were pseudo-randomly interleaved on the right or left of screen center. Feedback was represented by the presentation (reward) or not (non-reward) of a large green euro image.

In experiment 1, when P<1, participants performed a minimum of 48 trials per block in the training phases (*forced* and *free*) and the test phases. For P = 1, participants performed a minimum of 24 trials for training phases (*forced* and *free*) and 48 trials for test phase. Subjects selected almost exclusively the correct 2nd-stage target only after ~12 trials during training (Fig 2D, right panel) and maintained this performance level during testing (Fig 2E, right panel), thus, shortening the train phase for this deterministic condition did not affect subjects' behavior and allowed us to reduce the overall duration of the session. The orders of the blocks were randomly interleaved. In experiments 2 and 3 they performed a minimum of 40 trials for each block. Here, subjects started by performing experiment 3 followed by experiment 2. This was to ensure that the value of *free* trials was not devalued by experiment 2 when performing experiment 3. In experiment 3, subjects always started by the block with no target swaps (incoherence = 0), and in experiment 2 by the block with equal extrinsic reward probability (equivalent to the block P = 0.75 of experiment 1). All the other blocks were randomly interleaved.

## Trial structure

During the training phase, for each trial, subjects experienced the 1st-stage where a fixation point appeared in the center of the screen for 500ms, followed by one of the first two targets of the different trial types (*forced* or *free*) for an additional 750ms, either to the left or right of the fixation point (~11˚ from the center of the screen on the horizontal axis, 3˚ wide). Immediately after, the first target was turned off and two fractal targets appeared at the same eccentricity than the first target to the left and right of the fixation point. The subjects could then choose by themselves or had to match the target (depending on the trial type) using a key press (left or right arrow keys for left and right targets, respectively). After their selection, a red V-shape appeared for about 1000ms above the selected target (trace epoch). Importantly, in experiment 3, the red V-shape that informed the subject about their 2nd- stage target selection was initially light red and turned on for 250ms above the actual fractal target selected by the subjects. It was then changed to dark red for 750ms (i.e., same color than experiments 1 and 2). So, if the trial was incoherent, after 250ms, the light red V-shape jumped and reappeared in dark red simultaneously over the other target on the other side of the screen also for 750ms. Finally, the fixation point was turned off and the outcome was displayed during 750ms before the next trial. For the test phase, the timing was equivalent except for the decision epoch related to the first stage where participants could choose their favorite trial type (*free* and *forced* targets positioned randomly, left or right) after 500ms of fixation point presentation. When a selection was made, the first target remained for 500ms, associated to a red V-shape over the selected

1$^{st}$-stage target–indicating their choice. The second stage started with a 500ms epoch where only the fixation point was presented on the screen, followed by the fractal target presentation. During the first and second action epochs, no time pressure was imposed on subjects to make their choice, but if they pressed one of the keys during the first 100ms after target presentation ('early press'), a large red cross was displayed in the center of the screen for 500ms and the trial was repeated.

## Computational modelling

We fitted individual subject data with variants of temporal-difference reinforcement learning (TDRL) models. All models maintained a look-up table of state-action value estimates ($Q(s, a)$) for each unique target and each action across all conditions within a particular experiment. State-action values were updated at each stage ($t \in \{1,2\}$) within a trial according to the prediction error measuring the discrepancy between obtained and expected outcomes:

$$\delta_t = r_{t+1} + Z(s_{t+1}, a_{t+1}) - Q(s_t, a_t)$$

where $r_{t+1} \in \{0,1\}$ indicates whether the subject received an extrinsic reward, and $Z(s_{t+1}, a_{t+1})$ represents the current estimate of the state-action value. The latter could take three possible forms:

$$Z(s_{t+1}, a_{t+1}) = \begin{cases} Q(s_{t+1}, a_{t+1}) & \text{one}-\text{step SARSA} \\ \max_{a'} Q(s_{t+1}, a') & \text{Q}-\text{learning} \\ \beta \cdot \max_{a'} Q(s_{t+1}, a') + (1 - \beta) \cdot \min_{a'} Q(s_{t+1}, a') & \beta-\text{pessimistic} \end{cases}$$

Although Q-learning and SARSA variants differ in whether they learn off- or on-policy, respectively, we treated both algorithms as optimistic. Q-learning is strictly optimistic by considering only the best future state-action value, whereas SARSA can be more or less optimistic depending on the sensitivity of the mapping from state-action value differences to behavioral policy. We compared Q-learning and SARSA with a third state-action value estimator that incorporates risk attitude through a weighted mixture of the best and worst future action values (Gaskett's $\beta$-pessimistic model [36]). As $\beta \rightarrow 1$ the pessimistic estimate of the current state-action value converges to Q-learning.

The prediction error was then used to update all state-action values according to:

$$Q(s_t, a_t) \leftarrow Q(s_t, a_t) + \alpha \cdot \delta_t$$

where $\alpha \in [0,1]$ represents the learning rate.

We tested whether a free choice bonus could explain choice preference by modifying the obtained reward as follows:

$$r_{t+1} = r_{t+1}^{\text{extrinsic}} + \rho$$

where $\rho \in (-\inf, +\inf)$ is a scalar parameter added to any extrinsic reward following any action taken following selection of the *free* option.

Free actions at each stage were generated using a softmax policy as follows:

$$\pi(s, a^1) = \frac{\exp{(Q(s, a^1)/\tau)}}{\exp{(Q(s, a^1)/\tau)} + \exp{(Q(s, a^2)/\tau)}}$$

where increasing the temperature, $\tau \in [0, +\inf)$, produces a softer probability distribution over actions. The *forced* option, on the other hand, always led to the same fixed action. We used a

softmax behavioral policy for all TDRL variants, and in the context of our task, the Q-learning and SARSA algorithms were often similar in explaining subject data, so we treated them together in the main text (Fig D in S1 Text).

We also tested the possibility that subjects exhibited tendencies to alternate or perseverate on free or forced selections. We implemented this using a stickiness parameter that modified the policy as follows:

$$\pi(s, a^1) = \frac{\exp\left[(Q(s, a^1) + \kappa \cdot C_t(a^1))/\tau\right]}{\exp\left[(Q(s, a^1) + \kappa \cdot C_t(a^1))/\tau\right] + \exp\left[(Q(s, a^2) + \kappa \cdot C_t(a^2))/\tau\right]}$$

where the $\kappa \in (-\inf, +\inf)$ parameter represents the subject's tendance to perseverate, and $C_t(a)$ is a binary indicator for when a is a 1$^{st}$-stage action and was the same as was chosen on the previous trial.

We independently combined the free parameters to produce a family of model fits for each subject. We allowed the learning rate ($\alpha$) and softmax temperature ($\tau$) to differ for each of the two stages in a trial. We therefore fitted a total of 48 models (3 estimates of current state-action value [SARSA, Q, $\beta$-pessimistic] × presence or absence of free choice bonus [$\rho$] × 2- vs 1-learning rate [$\alpha$] × 2- vs 1-temperature [$\tau$] × presence or absence of stickiness [$\kappa$]).

## Parameter estimation and model comparison

We fitted model parameters using maximum a posteriori (MAP) estimation using the following priors:

$$\alpha \sim \text{beta}(\text{shape1} = 1.1, \ \text{shape2} = 1.1)$$

$$1/\tau \sim \text{gamma}(\text{shape} = 1.2, \ \text{scale} = 5)$$

$$\beta \sim \text{beta}(\text{shape1} = 1.1, \ \text{shape2} = 1.1)$$

$$\rho \sim \text{norm}(\text{mean} = 0, \ \text{sd} = 1)$$

$$\kappa \sim \text{norm}(\text{mean} = 0, \ \text{sd} = 1).$$

We based hyperparameters for $\alpha$ and $1/\tau$ on Daw and colleagues [42]. We used the same priors and hyperparameters for all models containing a particular parameter. We used limited-memory quasi-Newton algorithm (L-BFGS-B) to numerically compute MAP estimates, with $\alpha$ and $\beta$ bounded between 0 and 1 and $1/\tau$ bounded below at 0. For each model, we selected the best MAP estimate from 10 random parameter initializations.

For each model for each subject, we fitted a single set of parameters to both training and test data across conditions. Data from the training phase consisted of 2$^{nd}$-stage actions and rewards, but we also presented subjects, during the 1$^{st}$-stage stage, with the *free* or *forced* cues corresponding to the condition being trained. Therefore, we fitted the TDRL models assuming that the state-action values associated with the 1$^{st}$-stage fractals also underwent learning during the training phase, and that these backups continued into the test phase, where subjects actually made 1$^{st}$-stage decisions. That is, we initialized the state-action values during the test phase with the final state-action values during the training phase.

We treated initial state-action values as a hyperparameter. We set all initial Q-values equal to 0 or 0.5 to capture initial indifference between choice options (this decision rule considers only Q-value differences). Most subjects were best fit with initial Q-values = 0, although some

subjects were better fit with initial Q-values = 0.5 (Fig I in S1 Text, Tables A-B in S1 Text). We therefore selected the best Q-value initialization for each subject.

We used Schwarz weights to compare models, which provides a measure of the strength of evidence in favor of one model over others and can be interpreted as the probability that a model is best in the Bayesian Information Criterion (BIC) sense [67]. We calculated weights for each model as:

$$w_i(\text{BIC}) = \frac{\exp\left(-\Delta_i(\text{BIC})/2\right)}{\sum_{k=1}^{K} \exp\left(-\Delta_k(\text{BIC})/2\right)}$$

so that $\sum w_i(\text{BIC}) = 1$. We selected the model with the maximal Schwarz weight for each subject.

In order to verify that we could discriminate different state-action value estimates and how accurately we could estimate parameters, we performed model and parameter recovery analyses on simulated datasets (Fig D in S1 Text).

Note that the $\beta$-pessimist is identical to a Q-learner when $\beta = 1$. Since the $\beta$-pessimistic model includes an extra parameter, it is additionally penalized when it produces the same prediction as the Q-learner (i.e., when $\beta = 1$), and would not be selected unless there was sufficient evidence for weighting the worst possible outcome.

## Statistical analyses

We performed all analyses using R version 4.0.5 [68] and the following open-source R packages: afex version 1.2–1 [69], brms version 2.18.0 [70], emmeans version 1.8.4–1 [71], lme4 version 1.1–31 [72], mgcv version 1.8–41 [73], tidyverse version 1.3.2 [74].

We used generalized linear mixed models (GLMM) to examine differences in choice behavior. When the model did not include relative trial-specific information (e.g., reward on the previous trial), we aggregated data to the block level. Otherwise, we used choice data at the trial level. We included random effects by subject for all models (random intercepts and random slopes for the variable manipulated in each experiment; maximal expected value, relative expected value, or incoherence for experiments 1, 2, and 3, respectively). We performed GLMM significance testing using likelihood-ratio tests. We used Wald tests for post-hoc comparisons, and we corrected for multiple comparisons using Tukey's method. We used generalized additive mixed models (GAMM) to examine choice behavior as a function of trial within a block. We obtained smooth estimates of choice behavior using penalized regression splines, with penalization that allowed smooths to be reduced to zero effect [73]. We included separate smooths by block. We performed GAMM significance testing using approximate Wald-like tests [75].

We used Bayesian mixed models to analyze reaction time (RT) data because the right-skewed RT data was best fit by a shifted lognormal distribution, which is difficult to fit with standard GLMM software. For the RT analyses, we removed trials with RTs < 0.15 seconds and > 5 seconds (representing ~1% of the data). For each experiment we fit a model sampling four independent chains with 4000 iterations each for a total of 16000 posterior samples (the first 1000 samples of each chain discarded as warmup). Chain convergence was assessed using Gelman-Rubin statistics (R-hat < 1.01; [76]). We estimated 95% credible intervals using the highest posterior density, and obtained $p$-values by converting the probability of direction, which represents the closest statistical equivalent to a frequentist $p$-value [77].

## Supporting information

**S1 Text. Contains details of model and parameter recovery, as well as details concerning choice bias versus free choice bonus, Q-value initialization, and supplementary figures and tables.**
(PDF)

## Acknowledgments

The authors are grateful to Karim N'Diaye, operational manager of the PRISME platform at the ICM for his valuable help during participant testing.

## Author Contributions

**Conceptualization:** Jérôme Munuera, Valérian Chambon, Brian Lau.

**Data curation:** Jérôme Munuera, Marta Ribes Agost, Adrien Kerebel, Brian Lau.

**Formal analysis:** Jérôme Munuera, Marta Ribes Agost, Adrien Kerebel, Brian Lau.

**Funding acquisition:** Jérôme Munuera, Valérian Chambon, Brian Lau.

**Investigation:** Jérôme Munuera, Marta Ribes Agost, Adrien Kerebel, Valérian Chambon, Brian Lau.

**Methodology:** Jérôme Munuera, David Bendetowicz, Valérian Chambon, Brian Lau.

**Project administration:** Jérôme Munuera, Valérian Chambon, Brian Lau.

**Software:** Jérôme Munuera, Brian Lau.

**Supervision:** Jérôme Munuera, Valérian Chambon, Brian Lau.

**Validation:** Jérôme Munuera, Valérian Chambon, Brian Lau.

**Visualization:** Jérôme Munuera, Marta Ribes Agost, David Bendetowicz, Adrien Kerebel, Valérian Chambon, Brian Lau.

**Writing – original draft:** Jérôme Munuera, Valérian Chambon, Brian Lau.

**Writing – review & editing:** Jérôme Munuera, David Bendetowicz, Valérian Chambon, Brian Lau.

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
