## [Decision Letter · Decision Letter 0]

16 Nov 2022

Dear Dr Lau,

Thank you very much for submitting your manuscript "Choice seeking is motivated by the intrinsic need for personal control" for consideration at PLOS Computational Biology.

As with all papers reviewed by the journal, your manuscript was reviewed by members of the editorial board and by several independent reviewers. In light of the reviews (below this email), we would like to invite the resubmission of a significantly-revised version that takes into account the reviewers' comments.

Note especially the comments by reviewers 2 and 3 regarding the interpretation of the data, comments that may, or may not, be addressable by a rewrite and further analysis. 

We cannot make any decision about publication until we have seen the revised manuscript and your response to the reviewers' comments. Your revised manuscript is also likely to be sent to reviewers for further evaluation.

Sincerely,

Ulrik R. Beierholm

Academic Editor

PLOS Computational Biology

Marieke van Vugt

Section Editor

PLOS Computational Biology

Reviewer's Responses to Questions

**Comments to the Authors:**

Reviewer #1: Title: Choice seeking is motivated by the intrinsic need for personal control

In this paper, the authors carefully investigate whether choice-seeking is intrinsically motivating. Across three experiments, they show that people have a preference for free choice, even when free choice does not have greater relative value than forced choice. The authors use computational modeling to explore this behavior further, and the winning variation of their model could explain individual differences in choice seeking. The paper explores a classic topic using a modern framework, and is likely of interest to a broad audience. The thoughtful experimental designs are a particular strength of this paper. I have some comments and questions below that I think it would be useful for the authors to clarify/address.

1. One of the primary strengths of this paper is the experimental design/behavioral analyses. Therefore, it will be especially important for the authors to make certain that the complicated task designs are clearly explained.

a. For example, while it is discussed in the main text, there is no explanation of what the red “V” is in the Figure 1 caption. In addition, it wasn’t clear to me the purpose of both the red “V” and the black arrows, which both seem to indicate participant (or computer) selection.

b. In Experiment 3, I assume the black arrows/red “V” on incoherent trials simply point to the participant’s perceived selection, but I don’t think this is ever mentioned in the text.

c. Were participants paid additional compensation depending on their task earnings/performance? This should be mentioned in the text regardless of whether they were or not.

d. On Line 602, the authors state, “Data from the training phase consisted of 2nd-stage actions and rewards, but we also presented subjects with the 1st-stage cues corresponding to the condition being trained (forced or free).” This is the first mention that 1st stage cues are shown during the 2nd stage choice phase. It would be helpful if this was clarified earlier in the manuscript (either in Figure 1, the Methods section, or both).

2. Figure 2D: It’s not clear to me why in the “High” panel, there’s only data for 24 trials. If this is simply a case of me missing the explanation in the text, then perhaps it would be helpful to reference the figure in the text where this is explained.

3. I appreciated the clever controllability manipulation in Experiment 3. The authors explain that the repetition of free choices across incoherent trials suggests that participants are trying to “regain control.” However, I would imagine that this manipulation could cause the participants some confusion. In other words, pressing a “left” button and getting a “right” response probably seems like a bug or an error, and they might just be repeating their actions to test whether they have actually lost control. Because there typically isn’t any uncertainty around an action like a key press and a response, I’m wondering whether participants reported being confused by this manipulation, and also whether there were RT differences for repeated choices after incoherent trials.

4. In the context of Experiment 3, the authors talk about manipulating “perceived controllability.” I think it is worth being cautious of the language here. “Perceived” controllability implies that participants both explicitly reported a lack of control and yet truly had control over the choice. Instead, it might be best to simply talk about this as a manipulation of the actual controllability (which is what I believe it is, if I understand the task correctly).

5. The authors mention that they pooled two subject groups for Experiments 1a and 1b and that they did not have any “substantial differences.” It is important to include the actual evidence that these two subject groups did not differ behaviorally.

6. Beginning on Line 312, the authors write: “Consistent with previous studies, subjects preferred the opportunity to make a choice when expected rewards were equal between terminal actions (P = 0.5).” It would be helpful to cite some of this previous work here.

7. The authors should consider including a table in the supplement with each model name, brief details of the model (i.e., its parameters), and model fit values.

Reviewer #2: In this article, the authors aimed to identify the features triggering the well-known preference for free choices over forced choices. The authors replicated this effect in a 2-step task with two arms at step 1 –one leading deterministically to a free choice at the second step, the other to a forced option. The second-step terminal cues were probabilistically related to a binary outcome with the probability P of getting a reward being the same for both the best free choice outcome and for the forced cue outcome. Importantly, the experiment was divided into two phases: a training phase allowing participants to learn the terminal option-reward contingencies followed by a testing phase measuring free choice preference above and beyond learning. The main results are that participants prefer free choices after learning despite the risk of making a bad decision in step 2 free choice –even with a risk is maximised with P = 1 (Experiment 1). This result rules out that exploration would explain free choice preferences. Yet, this preference is reversed when the expected value of the best free choice option is much lower than the expected value of the forced option (Experiment 2). The authors offer a model-based mechanism posing that free choice preferences emerges from a bias placed on each free choice reward in a reinforcement learning model. Finally, they claim that free choice preference depends on personal control experience when the task is controllable.

Overall, this paper is well-written and I found it very interesting and exciting as free choice and more globally intrinsic motivation seem to be ubiquitous in everyday life and yet remains very poorly understood. It is also a growing topic of interest in the computational and cognitive neuroscience community. Here, the authors carefully designed experiments to test for various factors potentially influencing free choice preference. Moreover, they offered a mechanistic account for free choice preference using a model-based approach that is carefully and purposefully driven, ruling out many potential motivations for free choice preference. Most of the results are informative and neat.

My main concern is that the paper seems to fail identify what is behind free choice preference. Although the authors rule out many potential explanations through experiment 1 and 2, the data seem to rule out their proposed explanation in experiment 3 –personal control. See below for more details.

1) Experiment 3 aims to show that free choice preference is due to personal control or at least that it is influenced by personal control. To test this idea, stage 2 choice controllability (i.e. how likely it is that the chosen option is indeed selected) is altered, with the prediction that free choice preference should decline as controllability decreases.

1a) Yet, this may not to be the case in the data (see Figure 5B and mostly figure S3B that would deserve to be kept in the main text rather than to be hidden in the supplementary material). In fact, the crucial test for the effect of controllability on free choice preference is not reported. Thefore, the following claim lack of statistical support and is currently not appropriate: “Increasing the incoherence of the 2nd-stage actions progressively reduced choice preference (block 2 and 3: 67% and 64% in favor of free respectively).”. Statistics should be reported and interpreted with the corresponding claim.

1b) Similarly, the following statement can remain as such only if statistics indeed support an overall decline in free choice preference as incoherence increases. It should be removed otherwise. LL 261-262 “We found that *the decline in choice preference* depended on the 1st-stage choice on the previous trial.”

I found the other arguments supporting the need for personal control explanation of free choice preference rather vague:

2) I do not see how to interpret that “as incoherence increased, subjects tended to stay more with the forced option, while maintaining a preference to repeat free selections” and I am not sure that it is necessary to show that the motivation for personal control is what drives free choice preference. I suggest to better explain how important it is for the demonstration or to move it in the supplementary material.

3) The principal remaining argument to support that free choice preference depends on choice controllability is rather indirect. For all controllability levels and at stage 2, participants were sensitive to the previous reward when the selected option was the chosen one (i.e. if the previous choice was coherent) but were not so when the selected option was not the chosen one (i.e. if the previous choice was incoherent).

3a) Comparing statistically the incoherent stay proportion rewarded versus unrewarded would be required for this pattern to be fully demonstrated (as the stay proportion for coherent trials maybe lower than the unrewarded one). It should not be significantly different.

3b) Even then, I do not quite see how it proves that free choice preferences (measured at stage 1) depends on personal control. Perhaps it is because of the bonus placed on the reward in the learning model. Yet, a model with a choice bias may equally capture free choice preference (see below fore more details on that point).

Without necessary evidence, abstract, results and discussion sections would need to be rephrased. I have other less problematic concerns and some suggestions which I hope will help improve this article.

4) The authors interpret the observed consistent free choice preferences over different reward probability values (.5,.75, 1) as evidence ruling out curiosity, exploration, variety-seeking and selection-based explanations. While I can see why it rules out curiosity and exploration, I am not sure regarding the other alternative explanations at least as described in the introduction.

4a) Regarding the selection-based explanations, it is not clear to me in the results section why cognitive dissonance would have reduced effect after learning when terminal reward probabilities are clearly different between each free choice options. It may be worth to unpack this explanation (as it is done in the discussion)

4b) Regarding variety seeking, I guess that people may keep checking for changes in the testing phase when the reward probability is <1 but not when it is 1 and that is why curiosity and exploration explanations would be ruled out when reward probabilities are very different. Yet, there is still potentially more variety following the free choice option (as one can choose the worst option) whatever the reward probabilities are. Can the authors clarify the link between the results and how this explanation is ruled out?

5) The model space the authors used missed a model that could equally capture free choice preference: a choice bias in the softmax. The current winning model assumes a bonus plugged to the reward which predicts that Pfree will be inflated with respect to Pforced due to the boost, and it would predict an increasing free choice preference as the number of learning trials increases. Yet, this is not testable in the current design precisely because the participants already learned when they had the choice between the step 2 free or forced choice. Currently, I do not see any data supporting the current best model. I suggest the authors to either add the model I suggest in the model space and stress that these are two possible mechanisms that cannot be distinguishable with the current experiment, or to run a new experiment to add the possibility for participants to make step-1 decisions at various learning stages.

6) Here is another alternative model I suggest to add to the model space. The current best model includes a bonus placed on rewards following a free choice. If it actually predicts an increasing free choice preference, and if learning still occurs in the testing phase as suggested LL 184-188 pp 10-11, then it would predict an increasing free choice preference over trials in the testing phase. Yet, free choice preferences are constant (as can be observed in Figures 2C middle and right and Figure 3C left). I suggest to decrease the bonus strength linearly or exponentially for example as follows: rho_0 + rho_1 * trial number, or something like : rho_0 .* [rho_1 .^(trial number)]./rho_1 to account for the emergence of free choice preferences, with a reduced influence with time.

7) It clearly appears that participant learned which free-choice option was the best in the Experiment 1 training phase. Yet, it is unclear what is learned: the actual probability or that one option is better than the other? In other words, were participants really learning P? If so, did they learned accurately P to the same extent for the forced and for the free choice condition or is P estimation inflated in free as predicted by the learning model with a boost placed on rewards? One way to check that would be to explicitly ask the participants for the probability of each cue to lead to a reward after the training phase. Was it measured in anyway?

8) Are participants aware that the free choice corresponds to a 1-arm bandit (option 1 reward probability = P; option 2 reward probability = 1-P) and not to a 2-arm bandit (option 1 reward probability = P; option 2 reward probability = P’)? A possibility is that participants may overestimate the worst outcome reward probability depending on their prior (e.g. neutral priors 0.5 placed on each Q value at trial 0) and given that they did not choose them a lot and therefore given that they did not update them a lot. As a consequence, the free choice expected-value maybe higher than the forced choice expected value. This could be tested by mean of model comparison (even when the objective reward probability was set to be 1). I think that it would require Q to be initialised at 0.5.

9) Why are Q initialised at 0? Does model comparison show that it is better than 0.5?

10) Results showing free choice preference are a weakened by the fact that the extrinsic rewards were actually not implemented. It may be that participants only prefer free choices over forced choices when the cost is very low. In the current experiment, a reward probability difference of 0.2 is enough to reverse preferences at the group level. Yet, if extrinsic rewards were real, a much lower difference (maybe no difference) would be enough to reverse the effect. Adding this condition would make the result stronger. Alternatively, I suggest the authors to stress that the results validity is limited by the lack of actual extrinsic rewards.

11) I would not frame Experiment 2 as a titration. In this experiment, Pforced is either the same as Pfree, (Pforced=Pfree=.75), either larger (Pforced = .85 > Pfree), or much larger (Pforced = .95). The group average when Pforced =.85 is indeed around .5. Yet, the variance is very high and the distribution seems rather bimodal with a very few participants being indifferent between free choice and forced choice. I suggest the authors to frame it as a reversal: when the difference between Pforced and Pfree is 0.20 the vas majority of participants reversed their preference compared to when Pfroced=Pfree.

12) Results regarding the free choice preference are convincing (e.g; Figure 2B). Yet, there is a lot of variance in the data. Do the authors have a clue about why some participants always prefer the forced choices? Are they the ones who learned faster in the training phase?

13) It is not really clear what tests were performed in the result section. I suggest to report an effect size and to specify the test (as it seems to be a likelihood-ratio test, I suppose that the chi2 value could be reported).

14) I found it hard to get information about the model comparison results. In particular, figure 4A is not so informative as the models differs in terms of free parameters number. It is unclear what is the winning model at the group level and by how much. Reporting a figure or a table would be informative to get a sense of the second-best model, etc. In fact, reporting each model frequency on top would also be informative.

15) Lines 250-257 p 14: “Choice preference was high (70%) in block 1 when coherence was not altered, similar to block 1 from experiment 2 where extrinsic reward was equal between free and forced options”. Can the authors report some statistics to support that claim?

16) I wonder how the optimistic learner can be fully distinguishable from the pessimistic one as the former is embedded in the later. Can the author keep the pessimistic model and compare it to a version with beta = 0 and to another version with beta = 1? That would allow the authors to show that whether the best model is more optimistic, more pessimistic, or in between.

17) I suggest to add lines between each participant dot across all condition in the figure similar to figure 2B to provide the reader with an idea about preferences stability.

18) How were the sample sizes determined?

Reviewer #3: In this paper, Munuera and colleagues report results from three experiments using a two-stage decision task that aims to computationally characterize peoples’ preference for making choices. In Experiment 1, they found that even after learning the reward probabilities associated with second-stage choice options, participants demonstrated a consistent preference for choosing to make their own second-stage choice, rather than being forced to choose a maximally rewarding option. In Experiment 2, they found that participants continued to prefer to make their own second-stage choices, even when the forced choice option was more rewarding. And in Experiment 3, they demonstrated that in less controllable environments, when actions did not deterministically lead to expected states, participants’ preference for making free choices was reduced.

Overall, I think the experiments nicely demonstrate people’s preference for making choices, but many prior studies (which the authors cite) have demonstrated similar effects. The coherence manipulation in Experiment 3 is interesting and more novel, but I found the results difficult to interpret and the explanation the authors put forth for the effects overly speculative. Below, I lay out my concerns in more detail.

I think the authors’ hypothesis that participants might be ‘seeking personal control’ in response to the incoherence of the environment is very interesting. However, I found it difficult to follow their reasoning for why the specific pattern of observed results supported this hypothesis. For example, it is not clear to me why participants’ were necessarily “seeking personal control” by repeating their target selections regardless of reward outcomes on incoherent trials. It seems like an equally (or more) compelling explanation for these results is that participants rationally discounted the outcomes of events that were not caused by their own choices.

In addition, it is unclear how the computational account put forth for the findings from Experiment 1 and 2 can be extended to account for the findings from Experiment 3. The authors found that the majority of subjects in Experiment 1 and 2 were best fit by an optimistic reinforcement learning model with a free choice bonus that simply inflated the value of rewards obtained after choice. However, simply inflating the value of the rewards obtained from chosen options does not seem like it can explain the findings from Experiment 3 — instead, the model would need to estimate the coherence of the environment, and/or the probability that rewards are the result of one’s own actions, and modulate the bonus accordingly. Thus, while I found Experiment 3 interesting, I was unsure how to interpret the findings within the framework set up by the rest of the paper.

I also wondered about alternative computational accounts for the findings from Experiments 1 and 2. Here, the authors implemented the enhanced valuation of free choice outcomes via a bonus added to the reward outcome. It seems like other computational mechanisms could also explain these effects. In particular, prior studies have found that different higher learning rates for positive vs. negative outcomes following free choice can explain the preference. Is this model distinguishable from the choice bonus model described here?

In addition, it seems as though an alternative account could be that making the choice is in and of itself rewarding. Rather than inflating the value of the outcome obtained through free choice, an alternative model could add a bonus directly to the first-stage free choice option. In other words, do participants learn more from free vs. forced choices or simply prefer them?

Though perhaps outside the scope of the present work, one way to address this question might be to have an additional phase of the task where participants must choose between the forced choice fractal and the high-value free choice fractal. Does the high-value free choice fractal remain more valuable to participants? Similarly, do participants have accurate explicit knowledge of the reward probabilities associated with the fractals, or are their estimates for those following free choice inflated?

A few minor concerns:

It does not seem reasonable to initialize the Q value estimates at 0, since the actual outcomes values range from 0 - 1. Initializing at .5 seems less biased.

While the description of the task in the methods section at the end of the article is very clear, certain information that was critical for interpreting the results was not presented upfront. In particular, it would be helpful to know how long the training phase was, whether participants were instructed about the relationship between the reward probabilities, and how many participants completed Experiment 1.

**Have the authors made all data and (if applicable) computational code underlying the findings in their manuscript fully available?**

Reviewer #1: **No: **The authors said they *would* make the data available upon publication, but it was not available at the time of review.

Reviewer #2: **No: **The authors claimed that the code will be made available but they were not at the time of this review.

Reviewer #3: Yes

PLOS authors have the option to publish the peer review history of their article (what does this mean?). If published, this will include your full peer review and any attached files.

Reviewer #1: No

Reviewer #2: No

Reviewer #3: No
---

## [Decision Letter · Decision Letter 1]

26 Apr 2023

Dear Dr Lau,

Thank you very much for submitting your manuscript "Choice seeking is motivated by the intrinsic need for personal control" for consideration at PLOS Computational Biology. As with all papers reviewed by the journal, your manuscript was reviewed by members of the editorial board and by several independent reviewers. The reviewers appreciated the attention to an important topic. Based on the reviews, we are likely to accept this manuscript for publication, providing that you modify the manuscript according to the review recommendations.

------------

As you will see below all reviewers were impressed with the amount of work that went into the revision of the paper. While reviewers 1 and 3 were happy with the current version reviewer 2 has some significant objections to the interpretation of the results.

I am interested in what your response is to the reviewer comments, although the response is not likely to be sent to the reviewer.

Upon re-reading the paper I tend to partly agree with the reviewer. The in-coherence effect on free/forced choice is not significant, and the interaction between previous trial and coherence level is harder to interpret. Unlike the reviewer however I am much more open to merely changing the interpretation, including softening the title as one way forward. 

Ulrik Beierholm, Academic Editor

---------

Sincerely,

Ulrik R. Beierholm

Academic Editor

PLOS Computational Biology

Marieke van Vugt

Section Editor

PLOS Computational Biology

Reviewer's Responses to Questions

**Comments to the Authors:**

Reviewer #1: The authors have gone above and beyond to address my previous comments and I have no further suggestions. In terms of adding Figure R1 (RT) into the supplement, I think it would be useful, as it helps bolster the claims of the paper. However, I will leave the decision to include it up to the authors and the editor given space constraints and other considerations.

Reviewer #2: While I acknowledge that the authors put efforts in addressing my (and very similar reviewer 3’s) concerns, I still do not see any data supporting the title and the conclusions.

What is straightforward in the paper is not new (experiment 1 & 2), and what is new is not straightforward (experiment 3). In particular, the direct test of the hypothesis that “Choice seeking is motivated by the intrinsic need for personal control” is that free choice preference should be altered by lower level of controllability. That would show that when people are in control in free choice, they choose free choice, but when they are no longer, they equally prefer forced choice. This test was not reported in the first version of the manuscript and is in fact not significant in the revised version.

While the fact that incoherent choices are repeated regardless the outcome can be interpreted as a compensatory mechanism to feel in control when incoherency is high, it can be equally interpreted differently. For example, people may be model based learner, but they may attribute positive outcomes to their credit and they would repeat the same action. (Moreover, I think that the parametric modulation of this effect remains to be tested: the higher the incoherence level, the stronger the compensation should be.) It is anyway not a direct proof that “Choice seeking is motivated by the intrinsic need for personal control”.

Overall, I can only see three options here: the title not being supported by the data, the paper cannot be published as such (hence rejected); at least a pre-registered experiment 3’ including more incoherent levels (e.g. up to 0.5) shows that free choice preference is altered when personal control is lower (so that the need for personal control can no longer be satisfied); the paper is reframed to focus on the interesting (but not directly related to personal control) experiment 3 result, which is no longer about free choice and personal control.

In addition, I have other suggestions:

It remains unclear to me how the learning model with a bonus differs from a model with a choice bias.

Can the authors report a qualitative signature of this model that a model with a bias could not capture?

Model comparison is often relevant, but may be biased.

On that note, model recovery results could be performed (but even then, qualitative model validations are crucial).

In the same vein, I cannot explain why a q value initialised at 0 is better. If it is added as a free parameter, is it indeed estimated to be 0?

Reviewer #3: The authors have done an excellent job responding thoroughly to my concerns as well as those raised by other reviewers. I believe the changes they made to the manuscript greatly improve its clarity, and the additional modeling analyses provide stronger support for their original conclusions. I believe this paper will make a valuable contribution to the literature and am happy to recommend its acceptance.

**Have the authors made all data and (if applicable) computational code underlying the findings in their manuscript fully available?**

Reviewer #1: Yes

Reviewer #2: None

Reviewer #3: None

PLOS authors have the option to publish the peer review history of their article (what does this mean?). If published, this will include your full peer review and any attached files.

Reviewer #1: No

Reviewer #2: No

Reviewer #3: No

Figure Files:

Data Requirements:

Reproducibility:

References:

---

## [Editor Report · Decision Letter 2]

20 Jun 2023

Dear Dr Lau,

We are pleased to inform you that your manuscript 'Intrinsic motivation for choice varies with individual risk attitudes and the controllability of the environment' has been provisionally accepted for publication in PLOS Computational Biology.

Best regards,

Ulrik R. Beierholm

Academic Editor

PLOS Computational Biology

Marieke van Vugt

Section Editor

PLOS Computational Biology

---

## [Editor Report · Acceptance letter]

1 Aug 2023

PCOMPBIOL-D-22-01331R2 

Intrinsic motivation for choice varies with individual risk attitudes and the controllability of the environment

Dear Dr Lau,

I am pleased to inform you that your manuscript has been formally accepted for publication in PLOS Computational Biology. Your manuscript is now with our production department and you will be notified of the publication date in due course.

With kind regards,

Judit Kozma
